# On the Convergence to a Global Solution of Shuffling-Type Gradient Algorithms

**Lam M. Nguyen**[*]
IBM Research
Thomas J. Watson Research Center
Yorktown Heights, NY, USA
`LamNguyen.MLTD@ibm.com`

**Trang H. Tran**[*]
School of ORIE
Cornell University
Ithaca, NY, USA
`htt27@cornell.edu`

## Abstract

Stochastic gradient descent (SGD) algorithm is the method of choice in many machine learning tasks thanks to its scalability and efficiency in dealing with large-scale problems. In this paper, we focus on the shuffling version of SGD which matches the mainstream practical heuristics. We show the convergence to a global solution of shuffling SGD for a class of non-convex functions under over-parameterized settings. Our analysis employs more relaxed non-convex assumptions than previous literature. Nevertheless, we maintain the desired computational complexity as shuffling SGD has achieved in the general convex setting.

## 1 Introduction

In the last decade, neural network-based models have shown great success in many machine learning applications such as natural language processing [Collobert and Weston, 2008, Goldberg et al., 2018], computer vision and pattern recognition [Goodfellow et al., 2014, He and Sun, 2015]. The training task of many learning models boils down to the following finite-sum minimization problem:

$$\min_{w \in \mathbb{R}^d} \left\{ F(w) := \frac{1}{n} \sum_{i=1}^{n} f(w; i) \right\}, \tag{1}$$

where $f(\cdot; i) : \mathbb{R}^d \to \mathbb{R}$ is smooth and possibly non-convex for $i \in [n] := \{1, \cdots, n\}$. Solving the empirical risk minimization (1) had been a difficult task for a long time due to the non-convexity and the complicated learning models. Later progress with stochastic gradient descent (SGD) and its variants [Robbins and Monro, 1951, Duchi et al., 2011, Kingma and Ba, 2014] have shown great performance in training deep neural networks. These stochastic first-order methods are favorable thanks to its scalability and efficiency in dealing with large-scale problems. At each iteration SGD samples an index $i$ uniformly from the set $\{1, \ldots, n\}$, and uses the individual gradient $\nabla f(\cdot; i)$ to update the weight.

While there has been much attention on the theoretical aspect of the traditional i.i.d. (independently identically distributed) version of SGD [Nemirovski et al., 2009, Ghadimi and Lan, 2013, Bottou et al., 2018], practical heuristics often use without-replacement data sampling schemes. Also known as shuffling sampling schemes, these methods generate some random or deterministic permutations of the index set $\{1, 2, \ldots, n\}$ and apply gradient updates using these permutation orders. Intuitively, a collection of such $n$ individual updates is a pass over all the data, or an epoch. One may choose to create a new random permutation at the beginning of each epoch (in Random Reshuffling scheme) or use a random permutation for every epoch (in Single Shuffling scheme). Alternatively, one may use a

---

[*]Equally contributed. Corresponding author: Lam M. Nguyen

37th Conference on Neural Information Processing Systems (NeurIPS 2023).

Incremental Gradient scheme with a fixed deterministic order of indices. In this paper, we use the term unified shuffling SGD for SGD method using *any* data permutations, which includes the three special schemes described above.

Although shuffling sampling schemes usually show a better empirical performance than SGD [Bottou, 2009], the theoretical guarantees for these schemes are often more limited than vanilla SGD version, due to the lack of statistical independence. Recent works have shown improvement in computational complexity for shuffling schemes over SGD in various settings [Gürbüzbalaban et al., 2019, Haochen and Sra, 2019, Safran and Shamir, 2020, Nagaraj et al., 2019, Nguyen et al., 2021, Mishchenko et al., 2020, Ahn et al., 2020]. In particular, in a general non-convex setting, shuffling sampling schemes improve the computational complexity in terms of $\hat{\varepsilon}$ for SGD from $\mathcal{O}\left(\sigma^2/\hat{\varepsilon}^2\right)$ to $\mathcal{O}\left(n\sigma/\hat{\varepsilon}^{3/2}\right)$, where $\sigma$ is the bounded variance constant [Ghadimi and Lan, 2013, Nguyen et al., 2021, Mishchenko et al., 2020][2]. We summarize the detailed literature for multiple settings later in Table 1.

While global convergence is a desirable property for neural network training, the non-convexity landscape of complex learning models leads to difficulties in finding the global minimizer. In addition, there is little to no work studying the convergence to a global solution of shuffling-type SGD algorithms for a general non-convex setting. The closest line of research investigates the Polyak-Lojasiewicz (PL) condition (a generalization of strong-convexity), which demonstrates similar convergence rates as the strongly convex rates for shuffling SGD methods [Haochen and Sra, 2019, Ahn et al., 2020, Nguyen et al., 2021]. In another direction, Gower et al. [2021] and Khaled and Richtárik [2020] investigates the global convergence for some class of non-convex functions, however for vanilla SGD method. Beznosikov and Takáč [2021] investigate a random shuffle version of variance reduction methods (e.g. SARAH algorithm Nguyen et al. [2017]), but this approach only can show convergence to stationary points. With a target on shuffling SGD methods and specific learning architectures, we come up with the central question of this paper:

*How can we establish the convergence to global solution for a class of non-convex functions using shuffling-type SGD algorithms? Can we exploit the structure of neural networks to achieve this goal?*

We answer this question affirmatively, and our contributions are summarized below.

**Contributions**.

- We investigate a new framework for the convergence of a shuffling-type gradient algorithm to a global solution. We consider a relaxed set of assumptions and discuss their relations with previous settings. We show that our average-PL inequality (Assumption 3) holds for a wide range of neural networks equipped with squared loss function.

- Our analysis generalizes the class function called star-$M$-smooth-convex. This class contains non-convex functions and is more general than the class of star-convex smooth functions with respect to the minimizer (in the over-parameterized settings). In addition, our analysis does not use any bounded gradient or bounded weight assumptions.

- We show the total complexity of $\mathcal{O}\left(\frac{n}{\hat{\varepsilon}^{3/2}}\right)$ for a class of non-convex functions to reach an $\hat{\varepsilon}$-accurate global solution. This result matches the same gradient complexity to a stationary point for unified shuffling methods in non-convex settings, however, we are able to show the convergence to a global minimizer.

## 1.1 Related Work

In recent years, there have been different approaches to investigate the global convergence for machine learning optimization. This includes a popular line of research that studies some specific neural networks and utilizes their architectures. The most early works show the global convergence of Gradient Descent (GD) for simple linear networks and two-layer networks [Brutzkus et al., 2018, Soudry et al., 2018, Arora et al., 2019, Du et al., 2019b]. These results are further extended to deep learning architectures [Allen-Zhu et al., 2019, Du et al., 2019a, Zou and Gu, 2019]. This line of research continues with Stochastic Gradient Descent (SGD) algorithm, which proves the global convergence of SGD for deep neural networks for some probability depending on the initialization process and the number of input data [Brutzkus et al., 2018, Allen-Zhu et al., 2019, Zou et al.,

---

[2]The computational complexity is the number of (individual) gradient computations needed to reach an $\hat{\varepsilon}$-accurate stationary point (i.e. a point $\hat{w} \in \mathbb{R}^d$ that satisfies $\|\nabla F(\hat{w})\|^2 \le \hat{\varepsilon}$.)

2018, Zou and Gu, 2019]. The common theme that appeared in most of these references is the over-parameterized setting, which means that the number of parameters in the network are excessively large [Brutzkus et al., 2018, Soudry et al., 2018, Allen-Zhu et al., 2019, Du et al., 2019a, Zou and Gu, 2019]. This fact is closely related to our setting, and we will discuss it throughout our paper.

**Polyak-Lojasiewicz (PL) condition and related assumptions.** An alternative approach is to investigate some conditions on the optimization problem that may guarantee global convergence. A popular assumption is the Polyak-Lojasiewicz (PL) inequality, a generalization of strong-convexity [Polyak, 1964, Karimi et al., 2016, Nesterov and Polyak, 2006]. Using this PL assumption, it can be shown that (stochastic) gradient descent achieves the same theoretical rate as in the strongly convex setting (i.e linear convergence for GD and sublinear convergence for SGD) [Karimi et al., 2016, De et al., 2017, Gower et al., 2021]. Recent works demonstrate similar results for shuffling type SGD [Haochen and Sra, 2019, Ahn et al., 2020, Nguyen et al., 2021], both for unified and randomized shuffling schemes. On the other hand, [Schmidt and Roux, 2013, Vaswani et al., 2019] propose to use a new assumption called the Strong Growth Condition (SGC) that controls the rates at which the stochastic gradients decay comparing to the full gradient. This condition implies that the stochastic gradients and their variances converge to zero at the optimum solution [Schmidt and Roux, 2013, Vaswani et al., 2019]. While the PL condition for $F$ implies that every stationary point of $F$ is also a global solution, the SGC implies that such a point is also a stationary point of every individual function. However, complicated models as deep feed-forward neural networks generally have non-optimal stationary points [Karimi et al., 2016]. Thus, these assumptions are somewhat strong for non-convex settings.

Although there are plenty of works investigating the PL condition for the objective function $F$ [De et al., 2017, Vaswani et al., 2019, Gower et al., 2021], not many materials devoted to study the PL inequality for the individual functions $f(\cdot; i)$. A recent work [Sankararaman et al., 2020] analyzes SGD with the specific notion of gradient confusion for over-parameterized settings where the individual functions satisfy PL condition. They show that the neighborhood where SGD converges linearly depends on the level of gradient confusion (i.e. how much the individual gradients are negatively correlated). Taking a different approach, we investigate the PL property for individual functions and further show that our condition holds for a general class of neural networks with quadratic loss.

**Over-paramaterized settings for neural networks.** Most of the modern learning architectures contain deep and large networks, where the number of parameters are often far more than the number of input data. This leads to the fact that the objective loss function is trained closer and closer to zero. Understandably, in such settings all the individual functions $f(\cdot; i)$ are minimized simultaneously at 0 and they share a common minimizer. This condition is called the interpolation property (see e.g. [Schmidt and Roux, 2013, Ma et al., 2018, Meng et al., 2020, Loizou et al., 2021]) and is studied well in the literature (see e.g. [Zhou et al., 2019, Gower et al., 2021]). For a comparison, functions satisfying the strong growth condition necessarily satisfy the interpolation property. This property implies zero variance of individual gradients at the global minimizer, which allows good behavior for SGD near the solution. In this work, we slightly change this assumption which requires a small variance up to some level of the threshold $\varepsilon$. Note that when letting $\varepsilon \to 0$, our assumption exactly recovers the interpolation property.

**Star-convexity and related conditions.** There have been many attentions to a class of structured non-convex functions called star-convex [Nesterov and Polyak, 2006, Lee and Valiant, 2016, Bjorck et al., 2021]. Star-convexity can be understood as convexity between an arbitrary point $w$ and the global minimizer $w_*$. The name star-convex comes from the fact that each sublevel set is star-shaped [Nesterov and Polyak, 2006, Lee and Valiant, 2016]. Zhou et al. [2019] shows that if SGD follows a star-convex path and there exists a common global minimizer for all component functions, then SGD converges to a global minimum.

In recent progress, Hinder et al. [2020] considers the class of quasar-convex functions, which further generalizes star-convexity. This property was introduced originally in [Hardt et al., 2018] under the name 'weakly quasi-convex', and investigated recently in literature [Hinder et al., 2020, Jin, 2020, Gower et al., 2021]. This class uses a parameter $\zeta \in (0, 1]$ to control the non-convexity of the function, where $\zeta = 1$ yeilds the star-convexity and $\zeta$ approaches 0 indicates more non-convexity [Hinder et al., 2020]. Intuitively, quasar-convex functions are unimodal on all lines that pass through a global minimizer. Gower et al. [2021] investigates the performance of SGD for smooth and quasar-

convex functions using an expected residual assumption (which is comparable to the interpolation property). They show a convergence rate of $\mathcal{O}(1/\sqrt{K})$ for i.i.d. sampling SGD with the number of total iterations $K$, which translates to the computational complexity of $\mathcal{O}\left(1/\hat{\varepsilon}^2\right)$. To the best of our knowledge, this paper is the first work studying the relaxation of star-convexity and global convergence for SGD with shuffling sample schemes, not for the i.i.d. version.

## 2 Theoretical Setting

We first present the shuffling-type gradient algorithm below. Our convergence results hold for any permutation of the training data $\{1, 2, \ldots, n\}$, including deterministic and random ones. Thus, our theoretical framework is general and applicable for any shuffling strategy, including Incremental Gradient, Single Shuffling, and Random Reshuffling.

---
**Algorithm 1** (Shuffling-Type Gradient Algorithm for Solving (1))

1: **Initialization:** Choose an initial point $\tilde{w}_0 \in \text{dom}\,(F)$.
2: **for** $t = 1, 2, \ldots, T$ **do**
3:     Set $w_0^{(t)} := \tilde{w}_{t-1}$;
4:     Generate any permutation $\pi^{(t)}$ of $[n]$ (either deterministic or random);
5:     **for** $i = 1, \ldots, n$ **do**
6:         Update $w_i^{(t)} := w_{i-1}^{(t)} - \eta_i^{(t)} \nabla f(w_{i-1}^{(t)}; \pi^{(t)}(i))$;
7:     **end for**
8:     Set $\tilde{w}_t := w_n^{(t)}$;
9: **end for**

---

We further specify the choice of learning rate $\eta_i^{(t)}$ in the detailed analysis. Now we proceed to describe the set of assumptions used in our paper.

**Assumption 1.** *Suppose that* $f_i^* := \min_{w \in \mathbb{R}^d} f(w; i) > -\infty$, $i \in \{1, \ldots, n\}$.

**Assumption 2.** *Suppose that* $f(\cdot; i)$ *is $L$-smooth for all* $i \in \{1, \ldots, n\}$, *i.e. there exists a constant* $L \in (0, +\infty)$ *such that:*

$$\|\nabla f(w; i) - \nabla f(w'; i)\| \leq L\|w - w'\|, \quad \forall w, w' \in \mathbb{R}^d. \tag{2}$$

Assumption 1 is required in any algorithm to guarantee the well-definedness of (1). In most applications, the component losses are bounded from below. By Assumption 2, the objective function $F$ is also $L$-smooth. This Lipschitz smoothness Assumption is widely used for gradient-type methods. In addition, we denote the minimum value of the objective function $F_* = \min_{w \in \mathbb{R}^d} F(w)$. It is worthy to note the following relationship between $F_*$ and the component minimum values:

$$F_* = \min_{w \in \mathbb{R}^d} F(w) = \frac{1}{n} \min_{w \in \mathbb{R}^d} \left( \sum_{i=1}^{n} f(w; i) \right) \geq \frac{1}{n} \sum_{i=1}^{n} \min_{w \in \mathbb{R}^d} (f(w; i)) = \frac{1}{n} \sum_{i=1}^{n} f_i^*. \tag{3}$$

We are interested in the case where the set of minimizers of $F$ is not empty. The equality $F_* = \frac{1}{n} \sum_{i=1}^{n} f_i^*$ attains if and only if a minimizer of $F$ is also the common minimizer for all component functions. This condition implies that the variance of individual functions is 0 at the common minimizer.

### 2.1 PL Condition for Component Functions

Now we are ready to discuss the Polyak-Lojasiewicz condition as follows.

**Definition 1** (Polyak-Lojasiewicz condition). *We say that $f$ satisfies Polyak-Lojasiewicz (PL) inequality for some constant $\mu > 0$ if*

$$\|\nabla f(w)\|^2 \geq 2\mu[f(w) - f^*], \quad \forall w \in \mathbb{R}^d, \tag{4}$$

*where* $f^* := \min_{w \in \mathbb{R}^d} f(w)$.

The PL condition for the objective function $F$ is sufficient to show a global convergence for (stochastic) gradient descent [Karimi et al., 2016, Nesterov and Polyak, 2006, Polyak, 1964]. It is well known that a function satisfying the PL condition is not necessarily convex [Karimi et al., 2016]. However, this assumption on $F$ is somewhat strong because it implies that every stationary point of $F$ is also a global minimizer. Our goal is to consider a class of non-convex function which is more relaxed than the PL condition on $F$, while still having the good global convergence properties. In this paper, we formulated an assumption called "average PL inequality", specifically for the finite sum setting:

**Assumption 3.** *Suppose that $f(\cdot; i)$ satisfies average PL inequality for some constant $\mu > 0$ such that*

$$\frac{1}{n} \sum_{i=1}^{n} \|\nabla f(w; i)\|^2 \geq 2\mu \frac{1}{n} \sum_{i=1}^{n} [f(w; i) - f_i^*], \quad \forall w \in \mathbb{R}^d. \tag{5}$$

*where $f_i^* := \min_{w \in \mathbb{R}^d} f(w; i)$.*

***Comparisons.*** Assumption 3 is weaker than assuming the PL inequality for every component function $f(\cdot; i)$. In general setting, Assumption 3 is not comparable to assuming the PL inequality for $F$. Formally, if $F$ satisfies PL the condition for some parameter $\tau > 0$, then we have:

$$2\tau [F(w) - F_*] \leq \|\nabla F(w)\|^2 \leq \frac{1}{n} \sum_{i=1}^{n} \|\nabla f(w; i)\|^2. \tag{6}$$

However, by equation (3) we have that $[F(w) - F_*] \leq \frac{1}{n} \sum_{i=1}^{n} [f(w; i) - f_i^*]$. Therefore, the PL inequality for each function $f(\cdot; i)$, cannot directly imply the PL condition on $F$ and vice versa.

In the interpolated setting where there is a common minimizer for all component function $f(\cdot; i)$, it can be shown that the PL condition on $F$ is stronger than our average PL assumption:

$$2\tau \frac{1}{n} \sum_{i=1}^{n} [f(w; i) - f_i^*] = 2\tau [F(w) - F_*] \leq \|\nabla F(w)\|^2 \leq \frac{1}{n} \sum_{i=1}^{n} \|\nabla f(w; i)\|^2.$$

On the other hand, our assumption cannot imply the PL inequality on $F$ unless we impose a strong relationship that upper bound the sum of individual squared gradients $\frac{1}{n} \sum_{i=1}^{n} \|\nabla f(w; i)\|^2$ in terms of the full squared gradient $\|\nabla F(w)\|^2$, for every $w \in \mathbb{R}^d$. For these reasons, the average PL Assumption 3 is arguably more reasonable than assuming the PL inequality for the objective function $F$. Moreover, we show that Assumption 3 holds for a general class of neural networks with a final bias layer and squared loss function. We have the following theorem.

**Theorem 1.** *Let $\{(x^{(i)}, y^{(i)})\}_{i=1}^{n}$ is a training data set where $x^{(i)} \in \mathbb{R}^m$ is the input data and $y^{(i)} \in \mathbb{R}^c$ is the output data for $i = 1, \ldots, n$. We consider an architecture $h(w; i)$ with $w$ be the vectorized weight and $h$ consists of a final bias layer $b$:*

$$h(w; i) = W^T z(\theta; i) + b,$$

*where $w = \textbf{vec}(\{\theta, W, b\})$ and $z(\theta; i)$ are some inner architectures, which can be chosen arbitrarily. Next, we consider the squared loss $f(w; i) = \frac{1}{2} \|h(w; i) - y^{(i)}\|^2$. Then*

$$\|\nabla f(w; i)\|^2 \geq 2[f(w; i) - f_i^*], \quad \forall w \in \mathbb{R}^d, \tag{7}$$

*where $f_i^* := \min_{w \in \mathbb{R}^d} f(w; i)$.*

Therefore, for this application, Assumption 3 holds with $\mu = 1$.

## 2.2 Small Variance at Global Solutions

In this section, we change the interpolation property in previous literature [Ma et al., 2018, Meng et al., 2020, Loizou et al., 2021] by a small threshold. For any global solution $w_*$ of $F$, let us define

$$\sigma_*^2 := \inf_{w_* \in \mathcal{W}_*} \left( \frac{1}{n} \sum_{i=1}^{n} \|\nabla f(w_*; i)\|^2 \right). \tag{8}$$

We can show that when there is a common minimizer for all component functions (i.e. when the equality $F_* = \frac{1}{n} \sum_{i=1}^{n} f_i^*$ holds), the best variance $\sigma_*^2$ is 0. It is sufficient for our Theorem to impose a $\mathcal{O}(\varepsilon)$-level upper bound on the variance $\sigma_*^2$:

**Assumption 4.** *Suppose that the best variance at $w_*$ is small, that is, for $\varepsilon > 0$*

$$\sigma_*^2 \leq P\varepsilon, \tag{9}$$

*for some $P > 0$.*

It is important to note that in current non-convex literature, Assumption 4 alone (or, assuming $\sigma_*^2 = 0$ alone) is not sufficient enough to guarantee a global convergence property for SGD. Typically, some other conditions on the good landscape of the loss function are needed to complement the over-parameterized setting. Thus, we have motivation to introduce our next assumption.

## 2.3 Generalized Star-Smooth-Convex Condition for Shuffling Type Algorithm

We introduce the definition of star-smooth-convex function as follows.

**Definition 2.** *The function $g$ is star-$M$-smooth-convex with respect to a reference point $\hat{w} \in \mathbb{R}^d$ if*

$$\|\nabla g(w) - \nabla g(\hat{w})\|^2 \leq M\langle \nabla g(w) - \nabla g(\hat{w}), w - \hat{w}\rangle, \quad \forall w \in \mathbb{R}^d. \tag{10}$$

It is well known that when $g$ is $L$-smooth and convex [Nesterov, 2004], we have the following general inequality for every $w, w' \in \mathbb{R}^d$:

$$\|\nabla g(w) - \nabla g(w')\|^2 \leq L\langle \nabla g(w) - \nabla g(w'), w - w'\rangle \tag{11}$$

Our class of star-smooth-convex function requires a similar inequality to hold only for the special point $w' = \hat{w}$. Interestingly, this is related to a class of star-convex functions, which satisfies the convex inequality for the minimizer $\hat{w}$:

$$\text{(star-convexity w.r.t } \hat{w}) \quad g(w) - g(\hat{w}) \leq \langle \nabla g(w), w - \hat{w}\rangle, \; \forall w \in \mathbb{R}^d, \tag{12}$$

This class of functions contains non-convex objective losses and is well studied in the literature (see e.g. [Zhou et al., 2019]). Our Lemma 1 shows that the class of star-smooth-convex function is broader than the class of $L$-smooth and star-convex functions. Therefore, our problem of interest is non-convex in general.

**Lemma 1.** *The function $g$ is star-$M$-smooth-convex with respect to $\hat{w}$ for some constant $M > 0$ if one of the two following conditions holds:*

1. *$g$ is $L$-smooth and convex.*

2. *$g$ is $L$-smooth and $g$ is star-convex with respect to $\hat{w}$.*

*Proof.* The first statement of Lemma 1 follows directly from equation (11). We have that $g$ is star-$M$-smooth-convex with respect to any reference point and $M = L$.

Now we proceed to the second statement. From the star-convex property of $g$ with respect to $\hat{w}$, we have

$$g(w) - g(\hat{w}) \leq \langle \nabla g(w), w - \hat{w}\rangle, \; \forall w \in \mathbb{R}^d,$$

and $\nabla g(\hat{w}) = 0$ since $\hat{w}$ is the global minimizer of $g$. On the other hand, $g$ is $L$-smooth and we have

$$g(\hat{w}) \leq g\left(w - \frac{1}{L}\nabla g(w)\right) \leq g(w) - \frac{1}{2L}\|\nabla g(w)\|^2,$$

which is equivalent to $\|\nabla g(w)\|^2 \leq 2L[g(w) - g(\hat{w})]$, $i \in [n]$. Since $\nabla g(\hat{w}) = 0$, $i \in [n]$, we have for $\forall w \in \mathbb{R}^d$

$$\|\nabla g(w) - \nabla g(\hat{w})\|^2 \leq 2L[g(w) - g(\hat{w})] \overset{(12)}{\leq} 2L\langle \nabla g(w) - \nabla g(\hat{w}), w - w_*\rangle.$$

This is a star-$M$-smooth-convex function as in Definition 2 with $M = 2L$. $\qquad\square$

For the analysis of shuffling type algorithm in this paper, we consider the general assumption called the *generalized star-smooth-convex condition for shuffling algorithms*:

**Assumption 5.** *Using Algorithm 1, let us assume that there exist some constants $M > 0$ and $N \geq 0$ such that at each epoch $t = 1, \ldots, T$, we have for $i = 1, \ldots, n$:*

$$\|\nabla f(w_{i-1}^{(t)}; \pi^{(t)}(i)) - \nabla f(w_*; \pi^{(t)}(i))\|^2 \leq M \langle \nabla f(w_{i-1}^{(t)}; \pi^{(t)}(i)) - \nabla f(w_*; \pi^{(t)}(i)), w_{i-1}^{(t)} - w_* \rangle$$
$$+ N \frac{1}{n} \sum_{i=1}^{n} \|w_i^{(t)} - w_0^{(t)}\|^2, \tag{13}$$

*where $w_*$ is a global solution of $F$.*

We note that it is sufficient for our analysis to assume the case $N = 0$, i.e. the individual function $f(\cdot; i)$ is star-$M$-smooth-convex with respect to $w_*$ for every $i = 1, \ldots, n$ as in Definition 2. In that case, the assumption does not depend on the algorithm progress.

Assumption 5 is more flexible than the star-$M$-smooth-convex one in (10) because an additional term $N \frac{1}{n} \sum_{i=1}^{n} \|w_i^{(t)} - w_0^{(t)}\|^2$ for some constant $N > 0$ allows for extra flexibility in our setting, where the right-hand side term $\langle \nabla f(w; i) - \nabla f(w_*; i), w - w_* \rangle$ could be negative for some $w \in \mathbb{R}^d$.

Note that we do not impose any assumptions on bounded weights or bounded gradients. Therefore, the term $\frac{1}{n} \sum_{i=1}^{n} \|w_i^{(t)} - w_0^{(t)}\|^2$ cannot be uniformly bounded by any universal constant.

## 3 New Framework for Convergence to a Global Solution

In this section, we present our theoretical results. Our Lemma 2 first provides a recursion to bound the squared distance term $\|\tilde{w}_t - w_*\|^2$:

**Lemma 2.** *Assume that Assumptions 1, 2, 3, and 5 hold. Let $\{\tilde{w}_t\}_{t=1}^{T}$ be the sequence generated by Algorithm 1 with $0 < \eta_t \leq \min\left\{\frac{n}{2M}, \frac{1}{2L}\right\}$. For every $\gamma > 0$ we have*

$$\|\tilde{w}_t - w_*\|^2 \leq \left(1 + C_1 \eta_t^3\right) \|\tilde{w}_{t-1} - w_*\|^2 + C_2 \eta_t \sigma_*^2 - C_3 \eta_t [F(\tilde{w}_{t-1}) - F_*]. \tag{14}$$

*where $w_*$ is a global solution of $F$, $F_* = \min_{w \in \mathbb{R}^d} F(w)$, and*

$$\begin{cases} C_1 = \frac{8L^2}{3} + \frac{14NL^2}{M} + \frac{4\gamma L^4}{6M}, \\ C_2 = \frac{2}{M} + 1 + \frac{5}{6L^2} + \frac{8N}{3ML^2} + \frac{5\gamma}{12M}, \\ C_3 = \frac{\gamma}{\gamma+1} \frac{\mu}{M}. \end{cases} \tag{15}$$

Rearranging the results of Lemma 2, we have

$$F(\tilde{w}_{t-1}) - F_* \leq \frac{1}{C_3}\left(\frac{1}{\eta_t} + C_1 \eta_t^2\right) \|\tilde{w}_{t-1} - w_*\|^2 - \frac{1}{C_3 \eta_t} \|\tilde{w}_t - w_*\|^2 + \frac{C_2}{C_3}\sigma_*^2. \tag{16}$$

Therefore, with an appropriate choice of learning rate that guarantee $\left(1/\eta_t + C_1 \eta_t^2\right) \leq 1/\eta_{t-1}$, we can unroll the recursion from Lemma 2. Thus we have our main result in the next Theorem.

**Theorem 2.** *Assume that Assumptions 1, 2, 3, and 5 hold. Let $\{\tilde{w}_t\}_{t=1}^{T}$ be the sequence generated by Algorithm 1 with the learning rate $\eta_i^{(t)} = \frac{\eta_t}{n}$ where $0 < \eta_t \leq \min\left\{\frac{n}{2M}, \frac{1}{2L}\right\}$.*

*Let the number of iterations $T = \frac{\lambda}{\varepsilon^{3/2}}$ for some $\lambda > 0$ and $\varepsilon > 0$. Constants $C_1$, $C_2$, and $C_3$ are defined in (15) for any $\gamma > 0$. We further define $K = 1 + C_1 D^3 \varepsilon^{3/2}$ and specify the learning rate $\eta_t = K \eta_{t-1} = K^t \eta_0$ and $\eta_0 = \frac{D\sqrt{\varepsilon}}{K \exp(\lambda C_1 D^3)}$ such that $\frac{D\sqrt{\varepsilon}}{K} \leq \min\left\{\frac{n}{2M}, \frac{1}{2L}\right\}$ for some constant $D > 0$. Then we have*

$$\frac{1}{T} \sum_{t=1}^{T} [F(\tilde{w}_{t-1}) - F_*] \leq \frac{K \exp(\lambda C_1 D^3)}{C_3 D \lambda} \|\tilde{w}_0 - w_*\|^2 \cdot \varepsilon + \frac{C_2}{C_3} \sigma_*^2, \tag{17}$$

*where $F_* = \min_{w \in \mathbb{R}^d} F(w)$ and $\sigma_*^2$ is defined in (8).*

Our analysis holds for arbitrarily constant values of the parameters $\gamma$, $\lambda$ and $D$. In addition, we show our current analysis for every shuffling scheme. An interesting research question arises: whether the convergence results can be improved if one chooses to analyze a randomized shuffling scheme in this framework. However, we leave that question to future works.

Using Assumption 4, we can show the total complexity of Algorithm 1 for our setting.

Table 1: Comparisons of computational complexity (the number of individual gradient evaluations) needed by SGD algorithm to reach an $\hat{\varepsilon}$-accurate solution $w$ that satisfies $F(w) - F(w_*) \leq \hat{\varepsilon}$ (or $\|\nabla F(w)\|^2 \leq \hat{\varepsilon}$ in the non-convex case).

| Settings | References | Complexity | Shuffling Schemes | Global Solution |
|---|---|---|---|---|
| Convex | Nemirovski et al. [2009], Shamir and Zhang [2013] [1] | $\mathcal{O}\left(\frac{\Delta_0^2 + G^2}{\hat{\varepsilon}^2}\right)$ | ✗ | ✓ |
| | Mishchenko et al. [2020], Nguyen et al. [2021] [2] | $\mathcal{O}\left(\frac{n}{\hat{\varepsilon}^{3/2}}\right)$ | ✓ | ✓ |
| PL condition | Nguyen et al. [2021] | $\tilde{\mathcal{O}}\left(\frac{n\sigma^2}{\hat{\varepsilon}^{1/2}}\right)$ | ✓ | ✓ |
| Star-convex related | Gower et al. [2021] [3] | $\mathcal{O}\left(\frac{1}{\hat{\varepsilon}^2}\right)$ | ✗ | ✓ |
| Non-convex | Ghadimi and Lan [2013] [5] | $\mathcal{O}\left(\frac{\sigma^2}{\hat{\varepsilon}^2}\right)$ | ✗ | ✗ |
| | Nguyen et al. [2021], Mishchenko et al. [2020] [5] | $\mathcal{O}\left(\frac{n\sigma}{\hat{\varepsilon}^{3/2}}\right)$ | ✓ | ✗ |
| *Our setting (non-convex)* | **This paper, Corollary 1**[4] | $\mathcal{O}\left(\frac{n(N \vee 1)^{3/2}}{\hat{\varepsilon}^{3/2}}\right)$ | ✓ | ✓ |

[0] We note that the assumptions in this table are not comparable and we only show the roughly complexity in terms of $\hat{\varepsilon}$. In addition, to make fair comparisons, we only report the complexity of unified shuffling schemes.
[1] Standard results for SGD in convex literature often use a different set of assumptions from the one in this paper (e.g. bounded domain that $\|w - w_*\|^2 \leq \Delta_0$ for each iterate $w$ and/or bounded gradient that $\mathbb{E}[\|\nabla f(w; i)\|] \leq G^2$). We report the corresponding complexity for a rough comparison.
[2] [Mishchenko et al., 2020] shows a bound for Incremental Gradient while [Nguyen et al., 2021] has a unified setting. We translate these results for unified shuffling schemes from these references to the convex setting.
[3] Since we cannot find a reference containing the convergence rate for vanilla SGD and star-convex functions, we adapt the reference Gower et al. [2021] here. This paper shows a result for $L$-smooth and quasar convex function with an additional Expected Residual (ER) assumption, which is weaker than assuming smoothness for $f(\cdot; i)$ and interpolation property. The star-convex results hold when the quasar-convex parameter is 1.
[4] Since we use a different set of assumptions than the other references, we only report the rough comparison in $n$, $N$ and $\hat{\varepsilon}$, where $N$ is the constant from Assumption 5 and $N \vee 1 = \max(N, 1)$. Note that $N = 0$ in the framework of star-smooth-convex function. In addition, we need $\sigma_*^2 = 0$ so that the complexity holds with arbitrary $\hat{\varepsilon}$. We explain the detailed complexity below and in the Appendix.
[5] Standard literature for SGD in non-convex setting assumes a bounded variance that $\mathbb{E}_i[\|f(w; i) - \nabla F(w)\|^2] \leq \sigma^2$, we report the rough comparison.

**Corollary 1.** *Suppose that the conditions in Theorem 2 and Assumption 4 hold. Choose $C_1 D \lambda = 1$ and $\varepsilon = \hat{\varepsilon}/G$ such that $0 < \hat{\varepsilon} \leq G$ with the constants*

$$G = \frac{2C_1 D^2 e}{C_3} \|\tilde{w}_0 - w_*\|^2 + \frac{C_2 P}{C_3}, \text{ where}$$

$$\begin{cases} C_1 = \frac{8L^2}{3} + \frac{14NL^2}{M} + \frac{4L^2}{3M}, \\ C_2 = \frac{2}{M} + 1 + \frac{5}{6L^2} + \frac{8N}{3ML^2} + \frac{5}{12ML}, \\ C_3 = \frac{1}{L^2+1}\frac{\mu}{M}. \end{cases}$$

*Then, the we need $T = \frac{\lambda G^{3/2}}{\hat{\varepsilon}^{3/2}}$ epochs to guarantee*

$$\min_{1 \leq t \leq T}[F(\tilde{w}_{t-1}) - F_*] \leq \frac{1}{T}\sum_{t=1}^{T}[F(\tilde{w}_{t-1}) - F_*] \leq \hat{\varepsilon}.$$

***Computational complexity.*** Our global convergence result in this Corollary holds for a fixed value of $\hat{\varepsilon}$ in Assumption 4. Thus, when $\varepsilon \to 0$, this assumption is equivalent to assuming $\sigma_*^2 = 0$. The total complexity of Corollary 1 is $\mathcal{O}\left(\frac{n}{\hat{\varepsilon}^{3/2}}\right)$. This rate matches the best known rate for unified sampling schemes for SGD in convex setting [Mishchenko et al., 2020, Nguyen et al., 2021]. However, our result holds for a broader class of functions that are possibly non-convex. Comparing to the non-convex setting, current literature [Mishchenko et al., 2020, Nguyen et al., 2021] also matches our rate to the order of $\hat{\varepsilon}$, however, we can only prove that SGD converges to a stationary point with a weaker criteria $\|\nabla F(w)\|^2 \leq \hat{\varepsilon}$ for general non-convex funtions. Table 1 shows these comparisons in various settings. Note that when using a randomized shuffling scheme, SGD often performs a better rate in terms of the data $n$ in various settings with and without (strongly) convexity. For example, in strongly convex and/or PL setting, the convergence rate of RR is $\tilde{\mathcal{O}}(\sqrt{n}/\sqrt{\hat{\varepsilon}})$, which is better than unified schemes with $\tilde{\mathcal{O}}(n/\sqrt{\hat{\varepsilon}})$ [Ahn et al., 2020]. However, for a fair comparison, we do not report these results in Table 1 as our theoretical analysis is derived for unified shuffling scheme.

If we further assume that $L, M, N > 1$, the detailed complexity with respect to these constants is

$$\mathcal{O}\left(\frac{L^4(M+N)^{3/2}}{\mu^{3/2}} \cdot \frac{n}{\hat{\varepsilon}^{3/2}}\right).$$

We present all the detailed proofs in the Appendix. Our theoretical framework is new and adapted to the finite-sum minimization problem. Moreover, it utilizes the choice of shuffling sample schemes to show a better complexity in terms of $\hat{\varepsilon}$ than the complexity of vanilla i.i.d. sampling scheme.

## 4 Numerical Experiments

In this section, we show some experiments for shuffling-type SGD algorithms to demonstrate our theoretical results of convergence to a global solution. Following the setting of Theorem 1, we consider the non-convex regression problem with squared loss function. We choose fully connected neural networks in our implementation. We experiment with different regression datasets: the Diabetes dataset from sklearn library [Efron et al., 2004, Pedregosa et al., 2011] with 442 samples in dimension 10; the Life expectancy dataset from WHO [Repository, 2016] with 1649 trainable data points and 19 features. In addition, we test with the California Housing data from StatLib repository [Repository, 1997, Pedregosa et al., 2011] with a training set of 16514 samples and 8 features.

For the small Diabetes dataset, we use the classic LeNet-300-100 model [LeCun et al., 1998]. For other larger datasets, we use similar fully connected neural networks with an additional starting layer of 900 neurons. We apply the randomized reshuffling scheme using PyTorch framework [Paszke et al., 2019]. This shuffling scheme is the common heuristic in training neural networks and is implemented in many deep learning platforms (e.g. TensorFlow, PyTorch, and Keras [Abadi et al., 2015, Paszke et al., 2019, Chollet et al., 2015]).

For each dataset $\{x_i, y_i\}$, we preprocess and modify the initial data lightly to guarantee the over-parameterized setting in our experiment. We do this by using a pre-training stage: firstly we use GD/SGD algorithm to find a weight $w$ that yields a sufficiently small value for the loss function (for Diabetes dataset we train to $10^{-8}$ and for other datasets we train to $10^{-2}$). Letting the input data $x_i$ be fixed, we change the label data to $\hat{y}_i$ such that the weight $w$ yields a small loss function $\mathcal{O}(\epsilon)$ for the optimization associated with data $\{x_i, \hat{y}_i\}$, and the distance between $\hat{y}_i$ and $y_i$ is small. Then the modified data is ready for the next stage. We summarize the data (after modification) in our experiments in Table 2.

Table 2: Datasets used in our experiments

| Data name | # Samples | # Features | Networks layers | Sources |
|---|---|---|---|---|
| Diabetes | 442 | 10 | 300-100 | Efron et al. [2004] |
| Life Expectancy | 1649 | 19 | 900-300-100 | Repository [2016] |
| California Housing | 16514 | 8 | 900-300-100 | Repository [1997] |

For each dataset, we first tune the step size using a coarse grid search $[0.0001, 0.001, 0.01, 0.1, 1]$ for 100 epochs. Then, for example, if 0.01 performs the best, we test the second grid search $[0.002, 0.005, 0.01, 0.02, 0.05]$ for 5000 epochs. Finally, we progress to the training stage with $10^6$ epochs and repeat that experiment for 5 random seeds. We report the average results with confidence intervals in Figure 1.

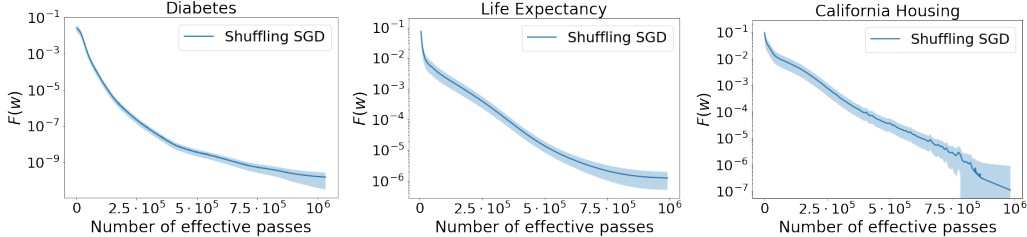

Figure 1: The train loss produced by Shuffling SGD algorithm for three datasets: Diabetes, Life Expectancy and California Housing.

For California Housing data, Shuffling SGD fluctuates toward the end of the training process. Nevertheless, for all three datasets it converges steadily to a small value of loss function. In summary, this experiment confirms our theoretical guarantee that demonstrates a convergence to global solution for shuffling-type SGD algorithm in neural network settings.

## 5    Conclusion and Future Research

In this paper, our focus is on investigating the global convergence properties of shuffling-type SGD methods. We consider a more relaxed set of assumptions in the framework of star-smooth-convex functions. We demonstrate that our approach achieves a total complexity of $\mathcal{O}\big(\frac{n}{\hat{\varepsilon}^{3/2}}\big)$ to attain an $\hat{\varepsilon}$-accurate global solution. Notably, this result aligns with the previous computational complexity of unified shuffling methods in non-convex settings, while ensuring that the algorithm converges to a global solution. Our theoretical framework revolves around the shuffling sample schemes for finite-sum minimization problems in machine learning.

We also provide insights into the relationships between our framework and well-established over-parameterized settings, as well as the existing literature on the star-convexity class of functions. Furthermore, we establish connections with neural network architectures and explore how these learning models align with our theoretical optimization frameworks.

This paper prompts several intriguing research questions, including practical network designs and more relaxed theoretical settings that can support the global convergence of shuffling SGD methods. Additionally, extending the global convergence framework to other stochastic gradient methods [Duchi et al., 2011, Kingma and Ba, 2014] and variance reduction methods [Le Roux et al., 2012, Defazio et al., 2014, Johnson and Zhang, 2013, Nguyen et al., 2017], all with shuffling sampling schemes, as well as the exploration of momentum shuffling methods [Tran et al., 2021, 2022], represents a promising direction.

An interesting research question that arises is whether the convergence results can be further enhanced by exploring the potential of a randomized shuffling scheme within this framework [Mishchenko et al., 2020]. However, we leave this question for future research endeavors.

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

# On the Convergence to a Global Solution of Shuffling-Type Gradient Algorithms Supplementary Material, NeurIPS 2023

## A    Theoretical settings: Proof of Theorem 1

### A.1    Proof of Theorem 1

*Proof.* Let us use the notation $f(w; i) = \phi_i(h(w; i)) = \frac{1}{2}\|h(w; i) - y^{(i)}\|^2$. We consider an architecture $h(w; i)$ with $w$ be the vectorized weight and $h$ consists of a final bias layer $b$:

$$h(w; i) = W^T z(\theta; i) + b \in \mathbb{R}^c,$$

where $w = \mathbf{vec}(\{\theta, W, b\})$ and $z(\theta; i)$ are some inner architecture, which can be chosen arbitrarily.

Firstly, we compute the gradient of $f(\cdot; i)$ with respect to $b \in \mathbb{R}^c$. For $j = 1, \ldots, c$, we have

$$\frac{\partial f(w; i)}{\partial b_j} = \frac{\partial \phi_i(h(w; i))}{\partial b_j} = \sum_{k=1}^{c} \frac{\partial h(w; i)_k}{\partial b_j} \cdot \frac{\partial \phi_i(x)}{\partial x_k}\Big|_{x=h(w;i)} = \frac{\partial \phi_i(x)}{\partial x_j}\Big|_{x=h(w;i)}, \ i = 1, \ldots, n. \tag{18}$$

The last equality follows since $\frac{\partial h(w;i)_k}{\partial b_j} = 0$ for every $k \neq j$ and $\frac{\partial h(w;i)_k}{\partial b_j} = 1$ for $k = j$. In other words, it is the identity matrix.

Let us denote that $f_i^* = \min_w f(w; i)$ and $\phi_i^* = \min_x \phi_i(x)$. We prove the following statement for $\mu = 1$:

$$\|\nabla_w f(w; i)\|^2 \geq \|\nabla_x \phi_i(x)|_{x=h(w;i)}\|^2 \geq 2\mu[\phi_i(h(w; i)) - \phi_i^*] \geq 2\mu[f(w; i) - f_i^*],$$

for every $w \in \mathbb{R}^d$, and $i = 1, \ldots, n$.

We begin with the first inequality:

$$\|\nabla_w f(w; i)\|^2 = \sum_{j=1}^{d} \left(\frac{\partial f(w; i)}{\partial w_j}\right)^2 \geq \sum_{j=d-c+1}^{d} \left(\frac{\partial f(w; i)}{\partial w_j}\right)^2 = \sum_{j=1}^{c} \left(\frac{\partial f(w; i)}{\partial b_j}\right)^2$$

$$\overset{(18)}{=} \sum_{j=1}^{c} \left(\frac{\partial \phi_i(x)}{\partial x_j}\Big|_{x=h(w;i)}\right)^2 = \|\nabla_x \phi_i(x)|_{x=h(w;i)}\|^2.$$

Now let us prove the PL condition for each function $\phi_i(x)$, i.e., there exists a constant $\mu > 0$ such that:

$$\|\nabla_x \phi_i(x)\|^2 \geq 2\mu[\phi_i(x) - \phi_i^*] \ \forall x \in \mathbb{R}^c, \ i = 1, \ldots, n.$$

Recall the squared loss that $\phi_i(x) = \frac{1}{2}\|x - y^{(i)}\|^2$ and $\nabla_x \phi_i(x) = x - y^{(i)}$. We can see that the constant $\mu = 1$ satisfies the following inequality for every $x \in \mathbb{R}^c, \ i = 1, \ldots, n$:

$$\|\nabla_x \phi_i(x)\|^2 = \|x - y^{(i)}\|^2 = 2\frac{1}{2}\|x - y^{(i)}\|^2 = 2\mu\phi_i(x) \geq 2\mu[\phi_i(x) - \phi_i^*],$$

where the last inequality follows since $\phi_i^* \geq 0$.

The PL condition for $\phi_i$ directly implies the second inequality. The last inequality follows from the facts that $f(w; i) = \phi_i(h(w; i))$ and $f_i^* = \min_w f_i \geq \min_x \phi_i(x) = \phi_i^*$. Hence, Theorem 1 is proved. □

# B  Preliminary results for SGD Shuffling Algorithm

In this section, we present the preliminary results for Algorithm 1. Firstly, from the choice of learning rate $\eta_i^{(t)} := \frac{\eta_t}{n}$ and the update $w_{i+1}^{(t)} := w_i^{(t)} - \eta_i^{(t)} \nabla f(w_i^{(t)}; \pi^{(t)}(i+1))$ in Algorithm 1, for $i \in [n]$, we have

$$w_i^{(t)} = w_{i-1}^{(t)} - \frac{\eta_t}{n} \nabla f(w_{i-1}^{(t)}; \pi^{(t)}(i)) = w_0^{(t)} - \frac{\eta_t}{n} \sum_{j=0}^{i-1} \nabla f(w_j^{(t)}; \pi^{(t)}(j+1)). \tag{19}$$

Hence,

$$w_0^{(t+1)} = w_n^{(t)} = w_0^{(t)} - \frac{\eta_t}{n} \sum_{j=0}^{n-1} \nabla f(w_j^{(t)}; \pi^{(t)}(j+1)). \tag{20}$$

Next, we refer to a Lemma in [Nguyen et al., 2021] to bound the updates of shuffling SGD algorithms.

**Lemma 3** (Lemma 5 in Nguyen et al. [2021]). *Suppose that Assumption 2 holds for* (1). *Let* $\{w_i^{(t)}\}$ *be generated by Algorithm 1 with the learning rate* $\eta_i^{(t)} := \frac{\eta_t}{n} > 0$ *for a given positive sequence* $\{\eta_t\}$. *If* $0 < \eta_t \le \frac{1}{2L}$ *for all* $t \ge 1$, *we have*

$$\frac{1}{n} \sum_{j=0}^{n-1} \|w_j^{(t)} - w_*\|^2 \le 4\|w_0^{(t)} - w_*\|^2 + 8\sigma_*^2 \cdot \eta_t^2, \tag{21}$$

$$\frac{1}{n} \sum_{j=0}^{n-1} \|w_j^{(t)} - w_0^{(t)}\|^2 \le \eta_t^2 \cdot \frac{8L^2}{3} \|w_0^{(t)} - w_*\|^2 + \frac{16L^2\sigma_*^2}{3} \cdot \eta_t^4 + 2\sigma_*^2 \cdot \eta_t^2. \tag{22}$$

Now considering the term $\|w_n^{(t)} - w_0^{(t)}\|^2$, we get that

$$\|w_n^{(t)} - w_0^{(t)}\|^2 \overset{(20)}{\le} \frac{\eta_t^2}{n} \left\| \frac{1}{n} \sum_{j=0}^{n-1} \nabla f(w_j^{(t)}; \pi^{(t)}(j+1)) \right\|^2$$

$$= \frac{\eta_t^2}{n} \left\| \frac{1}{n} \sum_{j=0}^{n-1} (\nabla f(w_j^{(t)}; \pi^{(t)}(j+1)) - \nabla f(w_*; \pi^{(t)}(j+1))) \right\|^2$$

$$\le \frac{\eta_t^2}{n} \frac{1}{n} \sum_{j=0}^{n-1} \left\| \nabla f(w_j^{(t)}; \pi^{(t)}(j+1)) - \nabla f(w_*; \pi^{(t)}(j+1)) \right\|^2$$

$$\overset{(2)}{\le} \frac{L^2\eta_t^2}{n} \frac{1}{n} \sum_{j=0}^{n-1} \|w_j^{(t)} - w_*\|^2$$

$$\overset{(21)}{\le} \frac{4L^2\eta_t^2}{n} \|w_0^{(t)} - w_*\|^2 + \frac{8L^2\eta_t^4}{n} \sigma_*^2.$$

We further have

$$\frac{1}{n} \sum_{j=0}^{n} \|w_j^{(t)} - w_0^{(t)}\|^2 = \frac{1}{n} \sum_{j=0}^{n-1} \|w_j^{(t)} - w_0^{(t)}\|^2 + \frac{1}{n}\|w_n^{(t)} - w_0^{(t)}\|^2$$

$$\le \eta_t^2 \cdot \frac{8L^2}{3}\|w_0^{(t)} - w_*\|^2 + \frac{16L^2\sigma_*^2}{3} \cdot \eta_t^4 + 2\sigma_*^2 \cdot \eta_t^2$$

$$+ \frac{4L^2\eta_t^2}{n}\|w_0^{(t)} - w_*\|^2 + \frac{8L^2\eta_t^4}{n}\sigma_*^2. \tag{23}$$

# C  Main results: Proofs of Lemma 4, Lemma 2, Theorem 2, and Corollary 1

## C.1  Proof of Lemma 4

**Lemma 4.** *Let $\{w_i^{(t)}\}_{t=1}^T$ be the sequence generated by Algorithm 1 with $\eta_i^{(t)} = \frac{\eta_t}{n}$, with $0 < \eta_t \leq \frac{n}{2M}$ for $\eta_t \leq \frac{1}{2L}$. Then, under Assumptions 1, 2, and 5, we have*

$$\|w_0^{(t+1)} - w_*\|^2 \leq \left(1 + B_1\eta_t^3\right)\|w_0^{(t)} - w_*\|^2 - \frac{\eta_t}{2M}\frac{1}{n}\sum_{i=1}^n \|\nabla f(w_{i-1}^{(t)}; \pi^{(t)}(i))\|^2 + B_2\eta_t\sigma_*^2,$$

(24)

*where*

$$\begin{cases} B_1 = \frac{8L^2}{3} + \frac{14NL^2}{M}, \\ B_2 = \frac{2}{M} + 1 + \frac{5}{6L^2} + \frac{8N}{3ML^2}. \end{cases}$$

(25)

*Proof.* We start with Assumption 5. Using the inequality $\frac{1}{2}\|a\|^2 - \|b\|^2 \leq \|a - b\|^2$, we have for $t = 1, \ldots, T$ and $i = 1, \ldots, n$:

$$\frac{1}{2}\|\nabla f(w_{i-1}^{(t)}; \pi^{(t)}(i))\|^2 - \|\nabla f(w_*; \pi^{(t)}(i))\|^2$$

$$\leq \|\nabla f(w_{i-1}^{(t)}; \pi^{(t)}(i)) - \nabla f(w_*; \pi^{(t)}(i))\|^2$$

$$\overset{(13)}{\leq} M\langle\nabla f(w_{i-1}^{(t)}; \pi^{(t)}(i)) - \nabla f(w_*; \pi^{(t)}(i)), w_{i-1}^{(t)} - w_*\rangle + N\frac{1}{n}\sum_{i=1}^n \|w_i^{(t)} - w_0^{(t)}\|^2$$

$$= M\langle\nabla f(w_{i-1}^{(t)}; \pi^{(t)}(i)), w_{i-1}^{(t)} - w_*\rangle - M\langle\nabla f(w_*; \pi^{(t)}(i)), w_{i-1}^{(t)} - w_*\rangle$$

$$+ N\frac{1}{n}\sum_{i=1}^n \|w_i^{(t)} - w_0^{(t)}\|^2,$$

This statement is equivalent to

$$-\langle\nabla f(w_{i-1}^{(t)}; \pi^{(t)}(i)), w_{i-1}^{(t)} - w_*\rangle \leq -\frac{1}{2M}\|\nabla f(w_{i-1}^{(t)}; \pi^{(t)}(i))\|^2 + \frac{1}{M}\|\nabla f(w_*; \pi^{(t)}(i))\|^2$$

$$- \langle\nabla f(w_*; \pi^{(t)}(i)), w_{i-1}^{(t)} - w_*\rangle$$

$$+ \frac{N}{M}\frac{1}{n}\sum_{i=1}^n \|w_i^{(t)} - w_0^{(t)}\|^2,$$

(26)

For any $w_* \in W^*$, from the update (19) we have,

$$\|w_i^{(t)} - w_*\|^2 \overset{(19)}{=} \|w_{i-1}^{(t)} - w_*\|^2 - \frac{2\eta_t}{n}\langle\nabla f(w_{i-1}^{(t)}; \pi^{(t)}(i)), w_{i-1}^{(t)} - w_*\rangle + \frac{\eta_t^2}{n^2}\|\nabla f(w_{i-1}^{(t)}; \pi^{(t)}(i))\|^2$$

$$\overset{(26)}{\leq} \|w_{i-1}^{(t)} - w_*\|^2 - \frac{2\eta_t}{2Mn}\|\nabla f(w_{i-1}^{(t)}; \pi^{(t)}(i))\|^2 + \frac{2\eta_t}{Mn}\|\nabla f(w_*; \pi^{(t)}(i))\|^2$$

$$- \frac{2\eta_t}{n}\langle\nabla f(w_*; \pi^{(t)}(i)), w_{i-1}^{(t)} - w_*\rangle + \frac{2\eta_t N}{Mn}\frac{1}{n}\sum_{i=1}^n \|w_i^{(t)} - w_0^{(t)}\|^2$$

$$+ \frac{\eta_t^2}{n^2}\|\nabla f(w_{i-1}^{(t)}; \pi^{(t)}(i))\|^2$$

$$\overset{(a)}{\leq} \|w_{i-1}^{(t)} - w_*\|^2 - \frac{\eta_t}{2Mn}\|\nabla f(w_{i-1}^{(t)}; \pi^{(t)}(i))\|^2 + \frac{2\eta_t}{Mn}\|\nabla f(w_*; \pi^{(t)}(i))\|^2$$

$$- \frac{2\eta_t}{n}\langle\nabla f(w_*; \pi^{(t)}(i)), w_{i-1}^{(t)} - w_*\rangle + \frac{2\eta_t N}{Mn}\frac{1}{n}\sum_{i=1}^n \|w_i^{(t)} - w_0^{(t)}\|^2$$

$$= \|w_{i-1}^{(t)} - w_*\|^2 - \frac{\eta_t}{2Mn}\|\nabla f(w_{i-1}^{(t)}; \pi^{(t)}(i))\|^2 + \frac{2\eta_t}{Mn}\|\nabla f(w_*; \pi^{(t)}(i))\|^2$$

$$- \frac{2\eta_t}{n}\langle\nabla f(w_*;\pi^{(t)}(i)), w_{i-1}^{(t)} - w_0^{(t)}\rangle - \frac{2\eta_t}{n}\langle\nabla f(w_*;\pi^{(t)}(i)), w_0^{(t)} - w_*\rangle$$

$$+ \frac{2\eta_t N}{Mn}\frac{1}{n}\sum_{i=1}^{n}\|w_i^{(t)} - w_0^{(t)}\|^2$$

$$\overset{(b)}{\leq} \|w_{i-1}^{(t)} - w_*\|^2 - \frac{\eta_t}{2Mn}\|\nabla f(w_{i-1}^{(t)};\pi^{(t)}(i))\|^2 + \frac{2\eta_t}{Mn}\|\nabla f(w_*;\pi^{(t)}(i))\|^2$$

$$+ \frac{\eta_t}{n}\|\nabla f(w_*;\pi^{(t)}(i))\|^2 + \frac{\eta_t}{n}\|w_{i-1}^{(t)} - w_0^{(t)}\|^2$$

$$- \frac{2\eta_t}{n}\langle\nabla f(w_*;\pi^{(t)}(i)), w_0^{(t)} - w_*\rangle + \frac{2\eta_t N}{Mn}\frac{1}{n}\sum_{i=1}^{n}\|w_i^{(t)} - w_0^{(t)}\|^2,$$

where $(a)$ follows since $\eta_t \leq \frac{n}{2M}$ and $(b)$ follows by the inequality $2\langle a, b\rangle \leq \|a\|^2\|b\|^2$.

Note that $\frac{1}{n}\sum_{i=1}^{n}\langle\nabla f(w_*;\pi^{(t)}(i)), w_0^{(t)} - w_*\rangle = \langle\nabla F(w_*), w_0^{(t)} - w_*\rangle = 0$ since $w_*$ is a global solution of $F$. Now we sum the derived statement for $i = 1, \ldots, n$ and get

$$\|w_n^{(t)} - w_*\|^2 \leq \|w_0^{(t)} - w_*\|^2 - \frac{\eta_t}{2M}\frac{1}{n}\sum_{i=1}^{n}\|\nabla f(w_{i-1}^{(t)};\pi^{(t)}(i))\|^2$$

$$+ \eta_t\left(\frac{2}{M} + 1\right)\frac{1}{n}\sum_{i=1}^{n}\|\nabla f(w_*;\pi^{(t)}(i))\|^2 + \frac{\eta_t}{n}\sum_{i=1}^{n}\|w_{i-1}^{(t)} - w_0^{(t)}\|^2$$

$$+ \frac{2N\eta_t}{M}\frac{1}{n}\sum_{i=1}^{n}\|w_i^{(t)} - w_0^{(t)}\|^2$$

$$\overset{(8),(22)}{\leq} \|w_0^{(t)} - w_*\|^2 - \frac{\eta_t}{2M}\frac{1}{n}\sum_{i=1}^{n}\|\nabla f(w_{i-1}^{(t)};\pi^{(t)}(i))\|^2$$

$$+ \left(\frac{2}{M} + 1\right)\eta_t\sigma_*^2 + \frac{8L^2\eta_t^3}{3}\|w_0^{(t)} - w_*\|^2 + \frac{16L^2\eta_t^5}{3}\sigma_*^2 + 2\eta_t^3\sigma_*^2$$

$$+ \frac{2N\eta_t}{M}\frac{1}{n}\sum_{i=1}^{n}\|w_i^{(t)} - w_0^{(t)}\|^2$$

$$\overset{(23)}{\leq} \|w_0^{(t)} - w_*\|^2 - \frac{\eta_t}{2M}\frac{1}{n}\sum_{i=1}^{n}\|\nabla f(w_{i-1}^{(t)};\pi^{(t)}(i))\|^2$$

$$+ \left(\frac{2}{M} + 1\right)\eta_t\sigma_*^2 + \frac{8L^2\eta_t^3}{3}\|w_0^{(t)} - w_*\|^2 + \frac{16L^2\eta_t^5}{3}\sigma_*^2 + 2\eta_t^3\sigma_*^2$$

$$+ \frac{16NL^2\eta_t^3}{3M}\|w_0^{(t)} - w_*\|^2 + \frac{32NL^2\eta_t^5}{3M}\sigma_*^2 + \frac{4N\eta_t^3}{M}\sigma_*^2$$

$$+ \frac{8NL^2\eta_t^3}{Mn}\|w_0^{(t)} - w_*\|^2 + \frac{16NL^2\eta_t^5}{Mn}\sigma_*^2,$$

where we apply the derivations from Lemma 3. Now noting that $\eta_t \leq \frac{1}{2L}$, $n \leq 1$ and rearranging the terms we get:

$$\|w_n^{(t)} - w_*\|^2 \leq \|w_0^{(t)} - w_*\|^2 - \frac{\eta_t}{2M}\frac{1}{n}\sum_{i=1}^{n}\|\nabla f(w_{i-1}^{(t)};\pi^{(t)}(i))\|^2$$

$$+ \left(\frac{8L^2}{3} + \frac{16NL^2}{3M} + \frac{8NL^2}{M}\right)\eta_t^3\|w_0^{(t)} - w_*\|^2$$

$$+ \left(\frac{2}{M} + 1 + \frac{1}{3L^2} + \frac{1}{2L^2} + \frac{2N}{3ML^2} + \frac{N}{ML^2} + \frac{N}{ML^2}\right)\eta_t\sigma_*^2$$

Since $w_n^{(t)} = w_0^{(t+1)} = \tilde{w}_t$, we have the desired result in (24). $\qquad\square$

## C.2 Proof of Lemma 2

*Proof.* From (24) where $B_1$ and $B_2$ are defined in (25), we have

$$\|w_0^{(t+1)} - w_*\|^2$$

$$\leq \left(1 + B_1\eta_t^3\right)\|w_0^{(t)} - w_*\|^2 - \frac{\eta_t}{2M}\frac{1}{n}\sum_{i=1}^n \|\nabla f(w_{i-1}^{(t)}; \pi^{(t)}(i))\|^2 + B_2\eta_t\sigma_*^2$$

$$\stackrel{(a)}{\leq} \left(1 + B_1\eta_t^3\right)\|w_0^{(t)} - w_*\|^2 - \frac{\gamma}{\gamma+1}\frac{\eta_t}{2M}\frac{1}{n}\sum_{i=1}^n \|\nabla f(w_0^{(t)}; \pi^{(t)}(i))\|^2$$

$$+ \frac{\eta_t\gamma}{2M}\frac{1}{n}\sum_{i=1}^n \|\nabla f(w_{i-1}^{(t)}; \pi^{(t)}(i)) - \nabla f(w_0^{(t)}; \pi^{(t)}(i))\|^2 + B_2\eta_t\sigma_*^2$$

$$\stackrel{(2)}{\leq} \left(1 + B_1\eta_t^3\right)\|w_0^{(t)} - w_*\|^2 - \frac{\gamma}{\gamma+1}\frac{\eta_t}{2M}\frac{1}{n}\sum_{i=1}^n \|\nabla f(w_0^{(t)}; \pi^{(t)}(i))\|^2$$

$$+ \frac{\eta_t\gamma L^2}{2M}\frac{1}{n}\sum_{i=1}^n \|w_{i-1}^{(t)} - w_0^{(t)}\|^2 + B_2\sigma_*^2$$

$$\stackrel{(22)}{\leq} \left(1 + B_1\eta_t^3\right)\|w_0^{(t)} - w_*\|^2 - \frac{\gamma}{\gamma+1}\frac{\eta_t}{2M}\frac{1}{n}\sum_{i=1}^n \|\nabla f(w_0^{(t)}; \pi^{(t)}(i))\|^2$$

$$+ \frac{\eta_t^3\gamma L^2}{2M}\left(\frac{8L^2}{3}\|w_0^{(t)} - w_*\|^2 + \frac{16L^2\sigma_*^2}{3}\cdot\eta_t^2 + 2\sigma_*^2\right) + B_2\eta_t\sigma_*^2$$

$$= \left(1 + B_1\eta_t^3 + \frac{4\eta_t^3\gamma L^4}{3M}\right)\|w_0^{(t)} - w_*\|^2 + \left(B_2 + \frac{\eta_t^2\gamma L^2}{M} + \frac{8\eta_t^4\gamma L^4}{3M}\right)\eta_t\sigma_*^2$$

$$- \frac{\gamma}{\gamma+1}\frac{\eta_t}{2M}\frac{1}{n}\sum_{i=1}^n \|\nabla f(w_0^{(t)}; \pi^{(t)}(i))\|^2$$

$$\stackrel{(5)}{\leq} \left(1 + \eta_t^3\left(B_1 + \frac{4\gamma L^4}{3M}\right)\right)\|w_0^{(t)} - w_*\|^2 + \left(B_2 + \frac{\eta_t^2\gamma L^2}{M} + \frac{8\eta_t^4\gamma L^4}{3M}\right)\eta_t\sigma_*^2$$

$$- \frac{\gamma}{\gamma+1}\frac{2\mu\eta_t}{2M}\frac{1}{n}\sum_{i=1}^n [f(w_0^{(t)}; \pi^{(t)}(i)) - f_i^*]$$

$$\stackrel{(3)}{\leq} \left(1 + \eta_t^3\left(B_1 + \frac{4\gamma L^4}{3M}\right)\right)\|w_0^{(t)} - w_*\|^2 + \left(B_2 + \frac{\eta_t^2\gamma L^2}{M} + \frac{8\eta_t^4\gamma L^4}{3M}\right)\eta_t\sigma_*^2$$

$$- \frac{\gamma}{\gamma+1}\frac{\mu\eta_t}{M}[F(w_0^{(t)}) - F_*]$$

$$\stackrel{(b)}{\leq} \left(1 + \eta_t^3\left(B_1 + \frac{4\gamma L^4}{3M}\right)\right)\|w_0^{(t)} - w_*\|^2 + \left(B_2 + \frac{\gamma}{4M} + \frac{\gamma}{6M}\right)\eta_t\sigma_*^2$$

$$- \frac{\gamma}{\gamma+1}\frac{\mu\eta_t}{M}[F(w_0^{(t)}) - F_*],$$

where $(a)$ follows since $-\|b\|^2 \leq \gamma\|a-b\|^2 - \frac{\gamma}{\gamma+1}\|a\|^2$ for any $\gamma > 0$ and $(b)$ follows since $\eta_t \leq \frac{1}{2L}$. Since $w_0^{(t+1)} = \tilde{w}_t$, we obtain the desired result in (14). $\square$

## C.3 Proof of Theorem 2

*Proof.* For $t = 1, \ldots, T = \frac{\lambda}{\varepsilon^{3/2}}$ for some $\lambda > 0$

$$\eta_t = (1 + C_1 D^3\varepsilon^{3/2})\eta_{t-1} = (1 + C_1 D^3\varepsilon^{3/2})^t\eta_0 \leq (1 + C_1 D^3\varepsilon^{3/2})^T\eta_0$$

$$= (1 + C_1 D^3\varepsilon^{3/2})^{\lambda/\varepsilon^{3/2}}\eta_0 = (1 + C_1 D^3\varepsilon^{3/2})^{\lambda/\varepsilon^{3/2}}\frac{D\sqrt{\varepsilon}}{(1 + C_1 D^3\varepsilon^{3/2})\exp(\lambda C_1 D^3)}$$

$$\leq \frac{D\sqrt{\varepsilon}}{(1 + C_1 D^3\varepsilon^{3/2})} \leq \min\left\{\frac{n}{2M}, \frac{1}{2L}\right\}, \tag{27}$$

since $(1+x)^{1/x} \le e$, $x > 0$. From (14), we have

$$[F(\tilde{w}_{t-1}) - F_*] \le \frac{1}{C_3}\left(\frac{1}{\eta_t} + C_1\eta_t^2\right)\|\tilde{w}_{t-1} - w_*\|^2 - \frac{1}{C_3\eta_t}\|\tilde{w}_t - w_*\|^2 + \frac{C_2}{C_3}\sigma_*^2. \tag{28}$$

We proceed to prove the following inequality for $t = 1, \ldots, T$,

$$\frac{1}{\eta_t} + C_1\eta_t^2 \le \frac{1}{\eta_{t-1}}. \tag{29}$$

From (27), and $\eta_t = K\eta_{t-1}$ where $K = (1 + C_1 D^3 \varepsilon^{3/2})$, we have

$$
\begin{aligned}
C_1\eta_t^2 = C_1 K^2 \eta_{t-1}^2 &= C_1 K^2 \frac{\eta_{t-1}^3}{\eta_{t-1}} \\
&\le C_1 K^2 \frac{D^3 \varepsilon^{3/2}}{K^3 \eta_{t-1}} = \frac{C_1 D^3 \varepsilon^{3/2}}{K\eta_{t-1}} && \text{since } \eta_{t-1} \le \frac{D\sqrt{\varepsilon}}{K} \\
&= \frac{K-1}{K}\frac{1}{\eta_{t-1}} = \frac{1}{\eta_{t-1}} - \frac{1}{K\eta_{t-1}} && \text{since } K = (1 + C_1 D^3 \varepsilon^{3/2}) \\
&= \frac{1}{\eta_{t-1}} - \frac{1}{K\eta_{t-1}} = \frac{1}{\eta_{t-1}} - \frac{1}{\eta_t}, && \text{since } \eta_t = K\eta_{t-1}.
\end{aligned}
$$

for $t = 1, \ldots, T$. Hence, from (28), we have

$$
\begin{aligned}
[F(\tilde{w}_{t-1}) - F_*] &\le \frac{1}{C_3}\left(\frac{1}{\eta_t} + C_1\eta_t^2\right)\|\tilde{w}_{t-1} - w_*\|^2 - \frac{1}{C_3\eta_t}\|\tilde{w}_t - w_*\|^2 + \frac{C_2}{C_3}\sigma_*^2 \\
&\le \frac{1}{C_3\eta_{t-1}}\|\tilde{w}_{t-1} - w_*\|^2 - \frac{1}{C_3\eta_t}\|\tilde{w}_t - w_*\|^2 + \frac{C_2}{C_3}\sigma_*^2.
\end{aligned}
$$

Averaging the statement above for $t = 1, \ldots, T$, we have

$$
\begin{aligned}
\frac{1}{T}\sum_{t=1}^{T}[F(\tilde{w}_{t-1}) - F_*] &\le \frac{1}{C_3\eta_0 T}\|\tilde{w}_0 - w_*\|^2 + \frac{C_2}{C_3}\sigma_*^2 \\
&\overset{(a)}{=} \frac{K\exp(\lambda C_1 D^3)}{C_3 D\sqrt{\varepsilon}}\frac{\varepsilon^{3/2}}{\lambda}\|\tilde{w}_0 - w_*\|^2 + \frac{C_2}{C_3}\sigma_*^2 \\
&= \frac{K\exp(\lambda C_1 D^3)}{C_3 D\lambda}\|\tilde{w}_0 - w_*\|^2 \cdot \varepsilon + \frac{C_2}{C_3}\sigma_*^2,
\end{aligned}
$$

where $(a)$ follows since $\eta_0 = \frac{D\sqrt{\varepsilon}}{K\exp(\lambda C_1 D^3)}$ and $T = \frac{\lambda}{\varepsilon^{3/2}}$. $\qquad\square$

### C.4 Proof of Corollary 1

*Proof.* Choose $\gamma = \frac{1}{L^2}$, we have

$$
\begin{cases}
C_1 = \frac{8L^2}{3} + \frac{14NL^2}{M} + \frac{4L^2}{3M}, \\
C_2 = \frac{2}{M} + 1 + \frac{5}{6L^2} + \frac{8N}{3ML^2} + \frac{5}{12ML}, \\
C_3 = \frac{1}{L^2+1}\frac{\mu}{M}.
\end{cases}
$$

Note that $K = 1 + C_1 D^3 \varepsilon^{3/2}$ and $C_1 D^3 = 1/\lambda$, we get that $K = 1 + 1/T \le 2$. We continue from the statement of Theorem 2 and the choice $C_1 D^3 \lambda = 1$:

$$
\begin{aligned}
\frac{1}{T}\sum_{t=1}^{T}[F(\tilde{w}_{t-1}) - F_*] &\le \frac{K\exp(\lambda C_1 D^3)}{C_3 D\lambda}\|\tilde{w}_0 - w_*\|^2 \cdot \varepsilon + \frac{C_2}{C_3}\sigma_*^2 \\
&\le \frac{2}{C_1 D^3 \lambda} \cdot \frac{C_1 D^2 \exp(\lambda C_1 D^3)}{C_3} \cdot \|\tilde{w}_0 - w_*\|^2 \cdot \varepsilon + \frac{C_2}{C_3}\sigma_*^2 && \text{since } K \le 2 \\
&\le \frac{2C_1 D^2 e}{C_3}\|\tilde{w}_0 - w_*\|^2 \cdot \varepsilon + \frac{C_2}{C_3}\sigma_*^2 && \text{since } C_1 D^3 \lambda = 1
\end{aligned}
$$

$$\leq \frac{2C_1 D^2 e}{C_3} \|\tilde{w}_0 - w_*\|^2 \cdot \varepsilon + \frac{C_2}{C_3} P\varepsilon \qquad \text{equation (9)}$$

$$\leq \left( \frac{2C_1 D^2 e}{C_3} \|\tilde{w}_0 - w_*\|^2 + \frac{C_2 P}{C_3} \right) \varepsilon = G\varepsilon$$

with

$$G = \frac{2C_1 D^2 e}{C_3} \|\tilde{w}_0 - w_*\|^2 + \frac{C_2 P}{C_3}.$$

Let $0 < \varepsilon \leq 1$ and choose $\hat{\varepsilon} = G\varepsilon$. Then the number of iterations $T$ is

$$T = \frac{\lambda}{\varepsilon^{3/2}} = \frac{\lambda G^{3/2}}{\hat{\varepsilon}^{3/2}}$$

$$= \frac{1}{\hat{\varepsilon}^{3/2} C_1 D^3} \left( \frac{2C_1 D^2 e \|\tilde{w}_0 - w_*\|^2 + C_2 P}{C_3} \right)^{3/2}$$

$$= \frac{1}{\hat{\varepsilon}^{3/2}} \cdot \frac{1}{C_1 D^3 C_3^{3/2}} \left( 2C_1 D^2 e \|\tilde{w}_0 - w_*\|^2 + C_2 P \right)^{3/2}$$

$$= \frac{1}{\hat{\varepsilon}^{3/2}} \cdot \frac{\left( 2D^2 e \|\tilde{w}_0 - w_*\|^2 \left( \frac{8L^2}{3} + \frac{14NL^2}{M} + \frac{4L^2}{3M} \right) + \left( \frac{2+M}{M} + \frac{5}{6L^2} + \frac{8N}{3ML^2} + \frac{5}{12ML} \right) P \right)^{3/2}}{\left( \frac{8L^2}{3} + \frac{14NL^2}{M} + \frac{4L^2}{3M} \right) D^3 \left( \frac{1}{L^2+1} \frac{\mu}{M} \right)^{3/2}}$$

to guarantee

$$\min_{1 \leq t \leq T} [F(\tilde{w}_{t-1}) - F_*] \leq \frac{1}{T} \sum_{t=1}^{T} [F(\tilde{w}_{t-1}) - F_*] \leq \hat{\varepsilon}.$$

Hence, the total complexity (number of individual gradient computations needed to reach $\hat{\varepsilon}$ accuracy) is $\mathcal{O}\left( \frac{n}{\hat{\varepsilon}^{3/2}} \right)$.

If we further assume that $L, M, N > 1$:

$$T = \frac{1}{\hat{\varepsilon}^{3/2}} \cdot \frac{\left( 2D^2 e \|\tilde{w}_0 - w_*\|^2 \left( \frac{8ML^2}{3} + 14NL^2 + \frac{4L^2}{3} \right) + \left( 2 + M + \frac{5M}{6L^2} + \frac{8N}{3L^2} + \frac{5}{12L} \right) P \right)^{3/2}}{\left( \frac{8L^2}{3} + \frac{14NL^2}{M} + \frac{4L^2}{3M} \right) D^3 \left( \frac{\mu}{L^2+1} \right)^{3/2}}$$

$$\leq \frac{1}{\hat{\varepsilon}^{3/2}} \cdot \left( \mathcal{O}((M+N)L^2) \right)^{3/2} \cdot \mathcal{O}\left( 1/L^2 \right) \cdot \left( \frac{L^2+1}{\mu} \right)^{3/2}$$

$$= \mathcal{O}\left( \frac{L^4 (M+N)^{3/2}}{\mu^{3/2}} \cdot \frac{1}{\hat{\varepsilon}^{3/2}} \right)$$

and the complexity is $\mathcal{O}\left( \frac{L^4 (M+N)^{3/2}}{\mu^{3/2}} \cdot \frac{n}{\hat{\varepsilon}^{3/2}} \right)$. $\qquad \square$

