In this paper, we study the global convergence property for shuffling-type SGD methods. We consider a relaxed set of assumptions in the framework of star-smooth-convex functions and show the total complexity of $\mathcal{O}\left(\frac{n}{\hat{\varepsilon}^{3/2}}\right)$ to reach an $\hat{\varepsilon}$-accurate global solution. This result matches the previous computational complexity of unified shuffling methods in convex settings. Our theoretical framework utilizes the choice of shuffling sample schemes for finite-sum minimization problems in machine learning. We provide discussions on the relations of our framework and the well-known over-parameterized settings, as well as current literature on the star-convexity class of functions. In addition, we show the connections to neural network architectures and discuss how these learning models fit into our optimization frameworks. Potential research questions arising from our paper include practical network designs and relaxed theoretical settings that support the global convergence of Shuffling SGD methods. Moreover, the global convergence framework for other stochastic gradient methods [Duchi et al., 2011, Kingma and Ba, 2014] and variance reduction methods [Nguyen et al., 2017, Beznosikov and Takáč, 2021] with shuffling sampling schemes is also an interesting direction.