# OpenReview forum: "On the Convergence to a Global Solution of Shuffling-Type Gradient Algorithms"
_NeurIPS.cc/2023/Conference — NeurIPS 2023 poster_

### Official Review · Reviewer_JrBJ · 2023-06-28

**Soundness:** 2 fair
**Presentation:** 2 fair
**Contribution:** 2 fair
**Rating:** 5
**Confidence:** 4

**Summary:**

This paper provides the convergence to a global solution of `shuffling' SGD (where the index set is permuted and apply gradient updates according to the permutation order) for a class of non-convex functions, in the over-parameterized setting.

**Strengths:**

The paper provides a good comprehensive background of previous work on the convergence of shuffling SGD. The theoretical results are stated clearly, and experiments on real data sets are implemented.

**Weaknesses:**

The big weakness of the paper is that the class of non-convex functions under consideration is not defined using interpretable or standard assumptions.  In particular, I am skeptical that this non-convex class is  weaker than previously considered function classes (in particular, the PL condition of Nguyen et al.)

The problem is that the quantities in Assumption 5 are not calculable beyond the convex setting;  only in the convex setting is it clear what the constant N in Assumption 5 is  (where N=0).  Beyond the convex setting, there is no way to measure N and in turn, gauge the sample complexity results presented in this paper.



**Questions:**

Please provide bounds on the constants in Assumption 5 for some concrete example classes of non-convex functions which were not included in previous global convergence proofs.





**Limitations:**

yes

---

> ### Author Rebuttal · Authors · 2023-08-10
>
> **Thank you for your constructive comments on our paper. We hope that our responses below answer all of your questions. Should you need any other clarification, please leave a comment and we are happy to discuss with you. While we are not able to upload a revision for the rebuttal, we will add all the details of our discussion to the next version of our paper.**
>
> > The big weakness of the paper is that the class of non-convex functions under consideration is not defined using interpretable or standard assumptions. In particular, I am skeptical that this non-convex class is weaker than previously considered function classes (in particular, the PL condition of Nguyen et al.)
>
> We would like to note that the PL condition on $F$ (in Nguyen et al): $|| \nabla F (w) ||^2 \geq 2 \mu [ F (w) - F_* ]$ is quite strong because it implies that every stationary point of the objective function $F$ is also a global minimizer ($|| \nabla F (w) ||^2 = 0 \Rightarrow F (w) - F_* = 0$), which is not true for most of the neural networks. The convergence criteria for non-convex is based on $|| \nabla F (w) ||^2 = 0$ so we only need to find a stationary point and automatically it is the global solution. For our set of assumptions, we cannot imply that a stationary point of the objective function $F$ is also a global minimizer.
>
> Moreover, we rigorously showed that our `average PL' condition (Assumption 3) holds for a wide class of neural networks with squared losses (in Theorem 1) -- so we do not even need to assume it since we already proved it.
>
> In addition, we would like to note that our analysis also holds under condition: $ || \nabla f(w_{i-1}^{(t)};\pi^{(t)} ( i )) - \nabla f(w_*;\pi^{(t)} ( i )) ||^2 \leq M \langle \nabla f(w_{i-1}^{(t)}; \pi^{(t)} ( i )) - \nabla f(w_*;\pi^{(t)} ( i )), w_{i-1}^{(t)} - w_* \rangle$ (star-M-smooth-convex in Definition 2) without needing the term $N \frac{1}{n} \sum_{i=1}^{n} || w_{i}^{(t)} - w_{0}^{(t)} ||^2$. Our Lemma 2 in the Appendix shows that the class of star-smooth-convex function (a class of non-convex functions) is broader than the class of $L$-smooth and star-convex functions -- non-convex class functions which were widely considered in the literature (see e.g. Zhou. et al, "SGD Converges to Global Minimum in Deep Learning via Star-convex Path", ICLR 2019). We only added the term $N \frac{1}{n} \sum_{i=1}^{n} || w_{i}^{(t)} - w_{0}^{(t)} ||^2$ since to allow extra flexibility for our setting. We can simply assume star-M-smooth-convex condition in Definition 2 without an extra term of $N \frac{1}{n} \sum_{i=1}^{n} || w_{i}^{(t)} - w_{0}^{(t)} ||^2$.
>
> For these reasons, we politely disagree with your impression that our assumptions are not interpretable. Since the PL assumption implies a global minimizer on stationary points, it might not be an effective tool to fully explain the behavior of training problems. On the other hand, our ‘average PL’ assumption is proved for a general class of neural networks with squared losses, which is a good way to interpret the assumption.
>
>
> > The problem is that the quantities in Assumption 5 are not calculable beyond the convex setting; only in the convex setting is it clear what the constant N in Assumption 5 is (where N=0). Beyond the convex setting, there is no way to measure N and in turn, gauge the sample complexity results presented in this paper and Please provide bounds on the constants in Assumption 5 for some concrete example classes of non-convex functions which were not included in previous global convergence proofs.
>
> Thank you for your comment. We believe this is a minor misunderstanding as we do not need to add the term $N \frac{1}{n} \sum_{i=1}^{n} || w_{i}^{(t)} - w_{0}^{(t)} ||^2$ for our analysis in order to obtain the desired result. We do this just to allow extra flexibility for our setting. Even without this term (i.e. $N = 0$), our class function is still non-convex since it is broader than the class of $L$-smooth and star-convex functions -- non-convex class functions which were widely considered in the literature (see e.g. Zhou. et al, "SGD Converges to Global Minimum in Deep Learning via Star-convex Path", ICLR 2019). We will consider removing the term with $N$ if you think it clarifies our paper better.
>
> **Finally, we hope our answers address all of your questions. Please do not hesitate to contact us if you need any additional clarifications regarding the assumptions of our paper. Although our assumption might not be standard and might not appear in prior work, we believe it offers novelty and interpretations to the setting of neural networks with squared loss function. We are looking forward to discussing these details with you. After our discussions, we kindly ask that you re-evaluate the paper and consider increasing your score.**

---

> ### Author Response · Authors · 2023-08-12
> **Follow up on the rebuttal**
>
> Dear Reviewer JrBJ,
>
> We hope our responses answer all your concerns. Please let us know if you have some time considering them. In case you need any remaining clarifications, we would be more than happy to discuss with you (within this discussion period). If your concerns are all properly addressed, we really hope that the reviewer positively re-evaluates our work.
>
> Regards,
>
> Authors

---

> > ### Comment · Reviewer_JrBJ · 2023-08-13
> >
> > Thank you for the detailed response.

---

> > > ### Author Response · Authors · 2023-08-13
> > > **Response to Reviewer JrBJ**
> > >
> > > Dear Reviewer JrBJ,
> > >
> > > We thank you very much for your reply. Could you please discuss with us whether you have any remaining concern? We are happy to answer your questions to help our paper's discussion. We would really appreciate it if you discuss with us early so that we can answer you thoroughly within this period.
> > >
> > > Thank you for your time and effort in reviewing this paper,
> > >
> > > Authors

---

> > > ### Author Response · Authors · 2023-08-15
> > > **Follow up**
> > >
> > > Dear Reviewer JrBJ,
> > >
> > > Since the Author-Reviewer discussion phase will end in a fews days, we would like to follow up and discuss with you. Please do not hesitate to contact us if there are additional answers or explanations that we can make to convince you of the significance of our paper within this discussion period. We appreciate your timely response, as it would provide us with an opportunity to address any remaining questions.
> > >
> > > We emphasize that our main contributions are in the form of a new developed theoretical framework to show the convergence to a global solution of shuffling type gradient algorithms. We have also provided some experiments to confirm our theory. Our theoretical results are significant since we are the first to obtain the convergence to a global solution for this class of non-convex problems. To the best of our knowledge, we are the first to provide the ‘average PL’ condition and show that it holds for a wide class of neural networks. Therefore, our theoretical results are significant: finding a global solution for a class of non-convex problems is an important and non-trivial problem.
> > >
> > > If your concerns are all properly addressed, we really hope that the reviewer positively re-evaluates our work to support this research direction. We appreciate your inputs and we thank you for your time spent reviewing this paper.
> > >
> > > Best regards,
> > >
> > > Authors

---

### Official Review · Reviewer_MeiV · 2023-07-05

**Soundness:** 4 excellent
**Presentation:** 3 good
**Contribution:** 2 fair
**Rating:** 6
**Confidence:** 3

**Summary:**

In this paper, the authors demonstrate the convergence of the shuffling version of Stochastic Gradient Descent (SGD) to a global solution for a class of non-convex functions. Notably, the analysis of the algorithm relies on more relaxed non-convex assumptions compared to previous works, while maintaining the same computational complexity as shuffling SGD in the convex setting.

**Strengths:**

1. The paper is clear and well written. Moreover, the motivation for the paper is well-explained.

 2. The authors assume more relaxed assumptions compared to previous work in the non-convex setting. The authors explain carefully the assumption and the contribution upon previous works.

3. By leveraging these assumptions, the authors derive a novel complexity bound for achieving convergence to a global minimizer. Notably, this complexity matches the performance of unified shuffling methods in previous works, which aimed for convergence to a stationary point.

**Weaknesses:**

A potential weakness of the paper is that the analysis, upon closer inspection, appears to follow a relatively straightforward path and bears similarities to previous works. In this context, the main distinguishing factor is the introduction of the average PL condition, which specifically enables the establishment of a bound for the shuffling version of SGD.

**Questions:**

In regards to Assumption 3, the authors claim that it is a weaker condition compared to assuming the PL inequality for every component function. However, when examining the example provided in Theorem 1 to satisfy Assumption 3, it appears that the presented example actually satisfies the PL inequality for every component function.
Is there are any well-known loss functions that fulfill Assumption 3 without necessarily satisfying the stronger condition of PL inequality for every component function?

**Limitations:**

yes

---

> ### Author Rebuttal · Authors · 2023-08-10
>
> **Thank you so much for your time and for your positive evaluations of our paper. We also thank you for your appreciation of our novel complexity bound. We hope that our responses below answer all of your questions. Should you need any other clarification, please leave a comment and we are happy to discuss with you. While we are not able to upload a revision for the rebuttal, we will add all the details of our discussion to the next version of our paper.**
>
>
> > A potential weakness of the paper is that the analysis, upon closer inspection, appears to follow a relatively straightforward path and bears similarities to previous works. In this context, the main distinguishing factor is the introduction of the average PL condition, which specifically enables the establishment of a bound for the shuffling version of SGD.
>
> We thank you for your comment! We believe that our proof techniques are novel and different from the previous works since we use different settings to obtain the convergence to a global solution while the previous work can only guarantee the convergence to a stationary point in the non-convex cases. Note that finding a global solution for a non-convex problem is very critical in optimization for machine learning.
>
>
> > In regards to Assumption 3, the authors claim that it is a weaker condition compared to assuming the PL inequality for every component function. However, when examining the example provided in Theorem 1 to satisfy Assumption 3, it appears that the presented example actually satisfies the PL inequality for every component function. Is there are any well-known loss functions that fulfill Assumption 3 without necessarily satisfying the stronger condition of PL inequality for every component function?
>
> We thank you for your insightful question! Indeed, Assumption 3 is weaker than the PL condition for every component function, and it is a strength of our analysis as it can deal with a more general assumption. While we are not aware of a well-known loss function that satisfies Assumption 3 without satisfying the component PL condition, it is an interesting research question. We note that since Theorem 1 already covers a broad class of neural networks with squared losses, it is understandable that the broader class may cover some of the machine learning losses. Thus the details of what other problems can be satisfied by our average PL assumption is a future research direction. This shows the potential of our work.
>
> **Finally, we hope our answers address all of your questions. We thank you again for your positive evaluations of our paper.  When all your concerns are resolved, we sincerely hope that you could consider increasing your score and support us.**

---

> ### Author Response · Authors · 2023-08-12
> **Follow up on the rebuttal**
>
> Dear Reviewer MeiV,
>
> We hope our responses answer all your questions!
>
> In case you need any remaining clarifications, we would be more than happy to reply. Please let us know your thoughts as soon as you can (within this discussion period). If your questions are all properly addressed, we really hope that you consider increasing your score to support our work.
>
> Regards,
>
> Authors

---

> > ### Comment · Reviewer_MeiV · 2023-08-17
> > **Thanks for the response!**
> >
> > Thanks to the authors for answering my questions!

---

### Official Review · Reviewer_7CVX · 2023-07-05

**Soundness:** 3 good
**Presentation:** 3 good
**Contribution:** 3 good
**Rating:** 6
**Confidence:** 3

**Summary:**

Shuffling-type SGD is widely used in practice, for which the prior works only provide gradient norm guarantees in nonconvex settings. This paper studies the convergence to a global solution for shuffling-type gradient algorithms under nonconvex settings, under a new analytical framework of more relaxed nonconvex assumptions, i.e., average PL condition and generalized star-smooth-convex condition.

**Strengths:**

1. The authors generalize two assumptions -- PL condition and star-M-smooth-convex condition -- to investigate the convergence to global solutions in nonconvex settings. Although average PL condition: cannot be directly compared with PL condition, in the interpolated settings (all component functions share the same minimizer) the average PL condition is weaker than PL condition and also covers certain neural networks with squared loss functions.
2. The authors provide a convergence guarantee of shuffling-type SGD towards global solutions under new assumptions, while the prior works only investigate either the global convergence for SGD sampling without replacement or the gradient norm guarantees for shuffling-type SGD.

**Weaknesses:**

1. I appreciate the discussion on the comparison between the average PL condition and PL condition the authors made in the main paper, and the applicability of average PL condition in neural networks. However, I am not sure about the comparison between these assumptions beyond the interpolated settings (which seems a strong and restricted assumption).
2. For the generalized star-M-smooth-convex condition (Assumption 5), I understand that the term $\frac{1}{n}\sum_{i = 1}^{n}||w_i^{(t)} - w_0^{(t)}||^2$ allows for more flexibility as the inner product term on the R.H.S. can be negative. But it still looks a bit unnatural to me, especially it depends on the all the iterates from the next (future) cycle generated by the algorithm.
3. Here is my major concern. The final result obtained in Theorem 2 with **increasing** learning rates only guarantees the convergence to global solutions within a fixed neighborhood of radius $O(\hat{\epsilon})$. The increasing learning rate does not match the empirical practice.

**Questions:**

1. The notation in Section 4 is confusing. The authors used $\epsilon$ in Assumption 4, but in Line 254 they say ``$\hat{\epsilon}$ in Assumption 4''. I feel that the authors are meant to use $\hat{\epsilon}$ for Assumption 4.
2. Could the authors comment on the comparison between the average PL condition and PL condition when $\sigma_*^2 \neq 0$?
2. Could the authors explain the reason of adding the term $\frac{1}{n}\sum_{i = 1}^{n}||w_i^{(t)} - w_0^{(t)}||^2$ in Assumption 5? Does it help the analysis? Can this term be changed to only dependent on $w_{i - 1}^{(t)}$? Because from the discussion in the paper, this term is added to allow negative $\langle \nabla f(w;i) - \nabla f(w_*; i), w - w_* \rangle$.
3. In prior works for shuffling-type SGD, the coefficient in front of $\sigma_*^2$ in the convergence bound like Eq. (17) in Theorem 2 usually depends on the stepsize $\eta_t$, then one can take $\eta_t = O(1/T)$ to guarantee exact convergence to the solutions. However, I see the constant coefficient for $\sigma_*^2$ in Eq. (17), and the authors are choosing increasing stepsizes as $\eta_t = K\eta_{t - 1}$ where $K > 1$ (meaning the initial stepsize needs to be very small). Could the authors explain the intuition for this stepsize choice and the reason not getting stepsize-dependent coefficient?

**Limitations:**

Please see the weaknesses and questions above.

---

> ### Author Rebuttal · Authors · 2023-08-10
>
> **Thank you so much for your time and your appreciation that the contribution and soundness of our paper is good. We hope that our responses below answer all of your questions. Should you need any other clarification, please leave a comment and we are happy to discuss with you. While we are not able to upload a revision for the rebuttal, we will add all the details of our discussion to the next version of our paper.**
> > I appreciate the discussion on the comparison between the average PL condition and PL condition the authors made in the main paper, and the applicability of average PL condition in neural networks. However, I am not sure about the comparison between these assumptions beyond the interpolated settings (which seems a strong and restricted assumption). and Could the authors comment on the comparison between the average PL condition and PL condition when $\sigma_{*}^2 \neq 0$?
>
> Beyond the interpolated settings, these two assumptions are not comparable. However, the PL condition on $F$: $|| \nabla F(w)||^2 \geq 2 \mu [F(w) - F_*]$ is quite strong because it implies that every stationary point of the objective function $F$ is also a global minimizer ($|| \nabla F(w)||^2=0 \Rightarrow F (w) - F_* = 0$), which is not true for most of the neural networks. The convergence criteria for non-convex is based on $|| \nabla F(w)||^2 = 0$ so it automatically implies the global solution if we could find a stationary point. For our set of assumptions, we cannot imply that a stationary point of the objective function $F$ is also a global minimizer. Moreover, we showed that our `average PL' condition holds for a wide class of neural networks with squared losses (in Theorem 1). Therefore, we believe that our average PL condition is more reasonable. In addition, over-parametrized setting is well-known in deep learning applications where the the number of parameters of a neural network is very large and could interpolate the data (see e.g. [Schmidt and Roux, 2013, Ma et al., 2018, Meng et al., 2020, Loizou et al., 2021]) (that is $\sigma_*^2 = 0$).
> > the term $\frac{1}{n} \sum_{i=1}^n ||w_i^{(t)}-w_0^{(t)}||^2$… looks a bit unnatural to me
>
> We would like to note that our analysis also holds under condition: $ ||\nabla f(w_{i-1}^{(t)};\pi^{(t)}(i))-\nabla f(w_*;\pi^{(t)}(i))||^2 \leq M \langle \nabla f(w_{i-1}^{(t)};\pi^{(t)} (i))-\nabla f(w_*;\pi^{(t)}(i)), w_{i-1}^{(t)}-w_* \rangle$ (star-M-smooth-convex in Definition 2) without needing the term $N \frac{1}{n}\sum_{i=1}^{n}||w_{i}^{(t)} - w_{0}^{(t)}||^2$. Our Lemma 2 in the Appendix shows that the class of star-smooth-convex function is broader than the class of $L$-smooth and star-convex functions -- non-convex class functions which were widely considered in the literature (see e.g. Zhou. et al, "SGD Converges to Global Minimum in Deep Learning via Star-convex Path", ICLR 2019). We added the term $N \frac{1}{n} \sum_{i=1}^{n}||w_{i}^{(t)}-w_{0}^{(t)}||^2$ just to allow extra flexibility for our setting. We can simply assume star-M-smooth-convex condition in Definition 2 without depending on the algorithm iterates and obtain the desired results. We will clarify this in the next version of our paper.
> > the increasing learning rate does not match the empirical practice.
>
> The increasing learning rates are not really an issue as long as they are still smaller than some threshold (in our case in Theorem 2: $0 < \eta_t \leq \min (\frac{n}{2M},\frac{1}{2L})$). There are some existing work with increasing learning rates, see e.g. [Tran-Dinh et al, A Hybrid Stochastic Optimization Framework for Composite Nonconvex Optimization, Math. Programming, 2022]. Although our guarantee for a global solution is within a fixed neighborhood of radius $\mathcal{0}(\hat{\epsilon})$, this $\hat{\epsilon}$ can be chosen to be small for some desired accuracy. Moreover, our learning rate is only increased by a small factor of $1+C*\epsilon^{3/2}$, which does not change the training process much. In fact, we have experimented with learning rate increasing by small factor (approximately 0.0001-0.000001) where the training trajectory is not much different from the constant learning rate. We will add the results in our revised version.
> > I feel that the authors are meant to use $\hat{\epsilon}$ for Assumption 4.
>
> We define $\epsilon=\hat{\epsilon}/G$ in Line 251 in Corollary 1. In Assumption 4, it is $\epsilon$.
> > Could the authors explain the reason of adding the term $\frac{1}{n} \sum_{i=1}^n||w_i^{(t)}-w_0^{(t)}||^2$ in Assumption 5? Does it help the analysis?
>
> As we mentioned above, we do not need to add the term $N \frac{1}{n} \sum_{i=1}^{n}||w_{i}^{(t)}-w_{0}^{(t)}||^2$ for our analysis in order to obtain the desired result. We do this just to allow extra flexibility for our setting.
> > Could the authors explain the intuition for this stepsize choice and the reason not getting stepsize-dependent coefficient?
>
> We thank you for your question! Our proof techniques are different from the previous works since we use different settings to obtain the convergence to a global solution while the previous work can only guarantee the convergence to a stationary point in the non-convex cases. Therefore, it is not strange if we could obtain a different bound including the constant coefficient for $\sigma_*^2$ as in Eq. (17). For this reason, we also need Assumption 4 (small $\sigma_*^2$ for overparametrized setting) to guarantee the $\epsilon$-accurate global solution. We choose a learning rate $\eta_t=K \eta_{t-1}$ in order to handle the recursive derivation in Eq. (28) in Appendix to obtain the desired result. Note that finding a global solution for a non-convex problem is very critical in optimization for machine learning.
>
> **Finally, we hope our answers address all of your questions. Since you appreciate the contributions of our paper, when all your concerns are resolved, we hope that you could re-evaluate the paper and consider increasing your score and support our approach.**

---

> ### Author Response · Authors · 2023-08-12
> **Follow up on the rebuttal**
>
> Dear Reviewer 7CVX,
>
> We hope our responses answer all your concerns. Please let us know if you have some time considering them. In case you need any remaining clarifications, we would be more than happy to discuss with you (within this discussion period). If your concerns are all properly addressed, we really hope that the reviewer positively re-evaluates our work.
>
> Regards,
>
> Authors

---

> > ### Comment · Reviewer_7CVX · 2023-08-13
> >
> > I thank the authors for their detailed responses, which address my questions. I am increasing my score to 6.

---

> > > ### Author Response · Authors · 2023-08-13
> > > **Thank you very much for your appreciation!**
> > >
> > > Dear Reviewer 7CVX,
> > >
> > > We thank you very much for your appreciation of our paper’s contributions and soundness. Your support means a lot to us and this will encourage us to pursue more significant work in this direction.
> > >
> > > Thank you again,
> > >
> > > Authors

---

### Official Review · Reviewer_VXbB · 2023-07-07

**Soundness:** 3 good
**Presentation:** 4 excellent
**Contribution:** 3 good
**Rating:** 6
**Confidence:** 3

**Summary:**

The paper analyzes convergence properties of shuffling-type methods (shuffle SGD and incremental gradient) to the global solution under relaxed assumptions for non-convex overparameterized functions. Particularly, the paper relaxes frequently used PL-condition to the averaged PL of local functions $f_i$, which is a weaker assumption under interpolation regime, and also imposes a weak star-smooth convex condition.

**Strengths:**

- The paper is well written and easy to follow. All the assumptions are well explained in the paper.
- Related works are well explained.
- I did not check the proof details but the convergence rate looks correct (although it seems it is possible to improve convergence rate for shuffle SGD variants).

**Weaknesses:**

1. My main concern is that the set of assumptions is not well motivated: would be nice to see examples of functions that do not satisfy PL-condition, but satisfy your relaxed conditions.

2. It is unclear how tight this bound is under proposed assumptions. Assuming PL, it is possible to achieve faster $\epsilon^{-1/2}$ convergence. Is it not possible to achieve under your set of assumptions ?

3. The paper advertises a lot analyzing shuffling-type algorithms, however its convergence rates are same as for the incremental gradient methods. For random reshuffling and single shuffling SGD faster dependence on the number of functions $n$ is possible to achieve.

4. I did not understand why in experiments the initial labels are changed. Wouldn’t it be enough just to take a large over-parameterized model ? Modifying the training data seems strange to me, especially because usually the purpose of the theory is to explain the realistic training scenarios. Coming up with new datasets that explain the theoretical results makes it look like the theory doesn’t apply to realistic training scenarios.

**Questions:**

1. Is assuming both Assumption 5 and Assumption 3 still weaker than just assuming PL (Def 1) ?

2. Is there a relation between a constant N and sigma ? (to compare the rates in Table 1).

3. Is the epsilon in line 241 the same as the epsilon in Assumption 4 ? If not, change the notation.

4. I did not find the definition of parameter $\lambda$ (used in line 241) in the paper.

5. Which variant of shuffling SGD was used in experiments ?

**Limitations:**

yes

---

> ### Author Rebuttal · Authors · 2023-08-10
>
> **Thank you so much for your time and your positive evaluations of our paper. We also thank you for your constructive comments and suggestions. We hope that our responses below answer all of your questions. Should you need any other clarification, please leave a comment and we are happy to discuss with you. While we are not able to upload a revision for the rebuttal, we will add all the details of our discussion to the next version.**
>
> > My main concern is that the set of assumptions is not well motivated: would be nice to see examples of functions that do not satisfy PL-condition, but satisfy your relaxed conditions.
>
> The traditional PL condition $|| \nabla F(w)||^2 \geq 2 \mu [F(w)-F_*]$ is quite strong because it implies that every stationary point of the objective function $F$ is also a global minimizer ($|| \nabla F(w)||^2=0 \Rightarrow F(w)-F_*=0$), which is not true for most of the neural networks. Hence we are motivated to consider weaker assumptions, e.g. we have proved that the PL condition on $F$ is stronger than our average PL assumption under over-parameterized settings (section 2.1, line 174). More importantly, we showed that our `average PL' condition holds for a wide class of neural networks with squared losses (in Theorem 1). Note that this general class of networks does not satisfy the PL-condition.
>
> > It is unclear how tight this bound is under proposed assumptions. Assuming PL, it is possible to achieve faster $\epsilon^{-1/2}$ convergence. Is it not possible to achieve under your set of assumptions?
>
> It is well-known that the property and behavior of the PL condition on $F$ is close to strongly convex conditions. Therefore, it is possible to obtain similar convergence rates for the PL condition on $F$ as in the strongly convex cases. As mentioned above, the PL condition on $F$ implies that every stationary point of the objective function $F$ is also a global minimizer, which is a strong assumption since the convergence criteria for non-convex problems is based on $|| \nabla F(w)||^2=0$ which automatically implies the global solution. For our set of assumptions, we cannot imply that a stationary point of the objective function $F$ is also a global minimizer. Therefore, our set of assumptions do not have such nice properties like the PL condition on $F$ or strongly convex, and the current results are the best we can do so far.
>
> > The paper advertises a lot analyzing shuffling-type algorithms, .... For random reshuffling and single shuffling SGD faster dependence on the number of functions $n$ is possible to achieve.
>
> The shuffling-type scheme can be understood as generating any permutation including incremental gradient methods, single shuffling, and random reshuffling. The goal of our analysis is focusing on the analysis for the general shuffling-type scheme. We thank you for your suggestion to improve the convergence bounds for random reshuffling schemes. This is an interesting question that we have considered before submission. However, showing improvement in the overall bound for RR is nontrivial since we are using different proof techniques compared to the previous work and we were only able to improve a factor of $n$ in one of the terms in the final bound. Although it may be possible to further improve the results, we strongly believe that the current contributions already merit publication. Due to the time limit, we leave the question of improving the rate for RR scheme to future work.
>
> > Modifying the training data seems strange to me, especially because usually the purpose of the theory is to explain the realistic training scenarios...
>
> Initially, we modified the training data in order to make sure that there exists a solution that interpolates the data. Thus this satisfies our purpose to show that the algorithm could find a global solution (and the modification on the data is not much). We agree with you that the alternative option is to find a large over-parameterized model, however, we cannot theoretically check how over-parameterized the model should be to guarantee that the interpolation exists, unless the loss function is trained to the minimizer in the experiment (which is the goal we are trying to demonstrate). We also note that our theory can only be partially/closely matched with realistic scenarios (as in our effort with Assumption 3), it can only match partially/approximately as our theory offers strong results (convergence to global solutions).
>
> > Is assuming both Assumption 5 and Assumption 3 still weaker than just assuming PL
>
> To the best of our knowledge, the two settings are not comparable. However, as mentioned above, the PL condition on $F$ has a very nice behavior since it implies that every stationary point of the objective function $F$ is also a global minimizer (which is a key for non-convex analysis since it just needs to find a stationary point to guarantee the global solution). We note that our assumptions do not have such property.
>
> > Is there a relation between a constant N and sigma
>
> $\sigma^2$ in Table 1 is a bounded variance constant while $N$ is a different constant that we introduce in this paper. They are from different assumptions so they don't have any relation.
>
> > Is the epsilon in line 241 the same as the epsilon in Assumption 4?
>
> $\epsilon$ in line 241 is the same as $\epsilon$ in Assumption 4.
>
> > I did not find the definition of parameter $\lambda$ (used in line 241) in the paper.
>
> The parameter $\lambda$ is a hyper-parameter that can be any constant greater than $0$. This $\lambda$ shows how the number of iterations of the algorithm compared to some order of epsilon.
> > Which variant of shuffling SGD was used in experiments?
>
> We use random reshuffling scheme for the experiments.
>
> **Finally, we hope our answers address all of your questions. We thank you again for your positive evaluations of our paper. When all your concerns are resolved, we sincerely hope that you could consider increasing your score and support us.**

---

> ### Author Response · Authors · 2023-08-12
> **Follow up on the rebuttal**
>
> Dear Reviewer VXbB,
>
> We hope our responses answer all your questions!
>
> In case you need any remaining clarifications, we would be more than happy to reply. Please let us know your thoughts as soon as you can (within this discussion period). If your questions are all properly addressed, we really hope that you consider increasing your score to support our work.
>
> Regards,
>
> Authors

---

> > ### Comment · Reviewer_VXbB · 2023-08-16
> >
> > I would like to thank the authors for their detailed replies to my questions. My main remaining concern is in justification of the class of functions considered.
> >
> > First, I have hoped for a more formal discussion why Example 1 is not PL.
> >
> > Second, since assuming Assumptions 3 & 5 is neither stronger, nor weaker than PL, but leads to a worse convergence rate, it is unclear why one should use Assumptions 3 & 5. Giving some important example that is covered by new assumptions but not covered by PL might solve this concern.

---

> > > ### Author Response · Authors · 2023-08-18
> > > **Reply to Reviewer VXbB**
> > >
> > > Dear Reviewer VXbB,
> > >
> > > Thank you for your insightful question! We are happy to discuss this in detail. Please see our answers below:
> > >
> > > > First, I have hoped for a more formal discussion why Example 1 is not PL.
> > >
> > > We can show this by a simple example (satisfying conditions in Theorem 1) as follows.
> > >
> > > Let $\{( {x}^{(i)},y^{(i)})\}_{i=1}^2$ be a training data set where $ {x}^{(i)} \in \mathbb{R}$ is the input data and $y^{(i)} \in \mathbb{R}$ is the  output data for $i = 1, 2$. Let us choose $z(\theta; i) = w_2 {x}^{(i)}$, so $h(w;i) =  w_1 (w_2 {x}^{(i)}) + b$, where $w = [ w_1 ; w_2 ; b ] \in \mathbb{R}^3$. (This is a small two-layer neural network).
> > >
> > > We have $f(w ; i) = \frac{1}{2} || h(w;i) - y^{(i)} ||^2 = \frac{1}{2} ( w_1 (w_2 {x}^{(i)}) + b - y^{(i)} )^2$ for $i = 1, 2$.
> > >
> > > Note that $F(w) = \frac{1}{n} \sum_{i=1}^n f(w ; i)$ and $\nabla F(w) = \frac{1}{n} \sum_{i=1}^n \nabla f(w ; i)$.
> > >
> > > We have $\nabla f (w; i) = [ w_2 {x}^{(i)} ( w_1 w_2 {x}^{(i)} + b - y^{(i)} ) ; w_1 {x}^{(i)} ( w_1 w_2 {x}^{(i)} + b - y^{(i)} ) ;  w_1 w_2 {x}^{(i)} + b - y^{(i)} ]$
> > >
> > > To demonstrate a specific case, we consider the data $x^{(1)} = 1$, $x^{(2)} = 0$, $y^{(1)} = 1$, $y^{(2)} = 2$. (Other data may work as well).
> > >
> > > We observe that, $w_* = [ 1 ; -1 ; 2 ]$ is the optimal solution of $F$ since $F_* = \frac{1}{2} \sum_{i=1}^2 \frac{1}{2} ( w_1^* (w_2^* {x}^{(i)}) + b^* - y^{(i)} )^2 = \frac{1}{2} \cdot \frac{1}{2} [(1 \cdot (-1) \cdot 1 + 2 - 1)^2 + (1 \cdot (-1) \cdot 0 + 2 - 2)^2] = 0$.
> > >
> > > On the other hand, there exists a stationary point $\hat{w} = [ 0 ; 0 ; 3/2 ]$ such that $\nabla F(\hat{w}) = \frac{1}{2} \sum_{i=1}^2 \nabla f(\hat{w} ; i) = \frac{1}{2} ( [ 0 \cdot 1 ( 0 \cdot 0 \cdot 1 + 3/2 - 1 ) ; 0 \cdot 1 ( 0 \cdot 0 \cdot 1 + 3/2 - 1 ) ;  0 \cdot 0 \cdot 1 + 3/2 - 1 ] + [ 0 \cdot 0 ( 0 \cdot 0 \cdot 0 + 3/2 - 2 ) ; 0 \cdot 0 ( 0 \cdot 0 \cdot 0 + 3/2 - 2 ) ;  0 \cdot 0 \cdot 0 + 3/2 - 2  ] ) = [ 0 ; 0 ; 0 ]$
> > >
> > > while $F(\hat{w}) = \frac{1}{2} \cdot \frac{1}{2} [ (0 \cdot 0 \cdot 1 + 3/2 - 1)^2 + (0 \cdot 0 \cdot 0 + 3/2 - 2)^2 ] = 0.125 \neq F_*$.
> > >
> > > Therefore, this example does not satisfy the PL condition on $F$ since it does not satisfy $|| \nabla F(\hat{w})||^2 \geq 2 \mu [F({\hat{w}})-F_*]$ for $\mu > 0$. Note that this is just an example to demonstrate that the general network in Theorem 1 does not necessarily satisfy the PL condition on $F$, but it satisfies our proposed Assumption 3.
> > >
> > > > Second, since assuming Assumptions 3 & 5 is neither stronger, nor weaker than PL, but leads to a worse convergence rate, it is unclear why one should use Assumptions 3 & 5. Giving some important example that is covered by new assumptions but not covered by PL might solve this concern.
> > >
> > > An example that helps to demonstrate our theory is the class of convex functions (but non-strongly convex). Note that the non-strongly convexity on $F$ cannot imply the PL condition on $F$. The example in Theorem 1 contains a class of non-strongly convex problem (e.g. $z (\theta ; i ) = x^{(i)}$), so the Assumptions 3 and 5 hold. While our theory obtains the same rate of convergences as the class of non-strongly convex problems, it does not restrict the problem to be convex. Thus, we believe our theoretical setting will be one of the directions to investigate non-convex problems with special structures.
> > >
> > > We hope our answers address all of your questions. Again, we are grateful for your positive assessment of our paper. We hope you find this direction is interesting and support us.
> > >
> > > Sincerely,
> > >
> > > Authors

---

> > > > ### Comment · Reviewer_VXbB · 2023-08-18
> > > >
> > > > I would like to thank the authors for their detailed response. I have no further questions, and I raise my score to 6.

---

> > > > > ### Author Response · Authors · 2023-08-18
> > > > > **Thank you!**
> > > > >
> > > > > Dear Reviewer VXbB,
> > > > >
> > > > > We thank you very much for your time and your appreciation. Your support is meaningful for us to explore this direction.
> > > > >
> > > > > Thank you again,
> > > > >
> > > > > Authors

---

### Author Rebuttal · Authors · 2023-08-10

**General Response:**

**First of all, we would like to thank the AC and all other reviewers for your hard work and for reviewing our paper.**

**To all reviewers:** Below we first highlight our contributions for you to have a better overview of our paper, after which we clarify current misunderstanding and/or confusion around our work. By addressing your concerns we hope you will be motivated to discuss our paper with us and clear up any remaining unclarity.

We emphasize that finding a global solution for a non-convex problem is very critical in optimization for machine learning. We further note that it is impossible to guarantee convergence to a global solution in a general non-convex setting without additional assumptions, and certain existing results rely on strong properties (e.g convexity, PL, strong growth) to obtain this. Our paper proposes another perspective of showing the convergence to a global solution by taking the advantage of the `average PL' condition which we show it holds for a wide class of neural networks. Overparametrized setting is a well-known setting in deep learning applications where the the number of parameters of a neural network is very large and could interpolate the data (see e.g. [Schmidt and Roux, 2013, Ma et al., 2018, Meng et al., 2020, Loizou et al., 2021]) (that is $\sigma_*^2 = 0$).

We emphasize that our main contributions are in the form of a new developed theoretical framework to show the convergence to a global solution of shuffling type gradient algorithms. We have also provided some experiments to confirm our theory. Our theoretical results are significant since we are the first to obtain the convergence to a global solution for this class of non-convex problems. To the best of our knowledge, we are the first to provide the 'average PL' condition and show that it holds for a wide class of neural networks. Therefore, our theoretical results are significant: finding a global solution for a class of non-convex problems is an important and non-trivial problem.

**This general response together with our individual responses to each reviewer address the reviewers' concerns. Given the significance of our theoretical and algorithmic framework which provides global convergence for non-convex problems, we sincerely encourage the reviewers to re-evaluate our work and consider increasing their scores.**

**Please do not hesitate to contact us if there are additional clarifications that we can make to convince the reviewers of the significance of our paper. We appreciate AC’s effort and time to handle the review discussion and we also value all the reviewers’ feedback and comments. We are looking forward to receiving follow-up responses and giving clarifications to help the discussion of our work.**

Sincerely,

Authors

---

### Author Response · Authors · 2023-08-21
**Thank you!**

Dear Area Chair and Reviewers,

We would like to thank the AC and the reviewers for your time and effort in handling and reviewing our paper.

Best regards,

Authors

---

### Decision · Program_Chairs · 2023-09-21

**Decision:**

Accept (poster)

**Comment:**

The paper analyzes the convergence of gradient-based optimizers under arbitrary orderings of the training data, also known as shuffling-type methods or incremental gradient. It analyzes convergence to the global solution under relaxed assumptions for non-convex over-parameterized objectives. Particularly, the paper relaxes frequently used PL-condition to the averaged PL of local functions, which is a weaker assumption under interpolation regime, and also imposes a weak star-smooth convex condition.

After discussion and feedback, the consensus is that this paper gives valuable contributions under interesting assumptions. The assumptions are particularly crucial as most literature uses assumptions far from realistic non-convex functions, so we positively evaluated the non-standard and more relaxed assumptions used here instead, although in some cases they could be explained more (e.g. specific examples showing the new assumptions are weaker).

Overall, the paper is well written, and we hope the detailed feedback can help to clarify the camera-ready version of the paper.